# Distillation Robustifies Unlearning

**Bruce W. Lee**[1,2][*][†]   **Addie Foote**[1][*]   **Alex Infanger**[1][*]
**Leni Shor**[1,3][*]   **Harish Kamath**[1][*]   **Jacob Goldman-Wetzler**[1,4][*]
**Bryce Woodworth**[1]   **Alex Cloud**[*]   **Alexander Matt Turner**
[1]ML Alignment & Theory Scholars   [2]University of Pennsylvania
[3]Massachusetts Institute of Technology   [4]Brown University

## Abstract

Current LLM unlearning methods are not robust. A few steps of finetuning can revert their effects. We begin by showing that this is true even for an idealized form of unlearning: training to imitate a model that was never trained on unwanted information. This shows that training a model can drastically modify its input-output behavior while leaving its underlying capabilities intact. In light of this dynamic, we show our main result. Training a randomly initialized student on the outputs of an unlearned model transfers behaviors while leaving latent capabilities behind. In short, *distillation robustifies unlearning*. Based on this result, we propose Unlearn-Noise-Distill-on-Outputs (UNDO), a scalable method that distills an unlearned model into a noised copy of itself. UNDO introduces a tunable tradeoff between compute cost and robustness, establishing a new Pareto frontier on synthetic language and arithmetic tasks. At its strongest setting, UNDO matches the robustness of a model retrained from scratch with perfect data filtering while using only 60-80% of the compute and requiring only 0.01% of the pretraining data to be labeled. We also show that UNDO robustifies unlearning on the more realistic Weapons of Mass Destruction Proxy (WMDP) benchmark. Since distillation is widely used in practice, incorporating an unlearning step beforehand offers a convenient path to robust capability removal.

## 1  Introduction

Large language models (LLMs) can acquire harmful capabilities during pretraining on massive, largely unfiltered datasets [9, 64]. This complicates model deployment. For example, a model with knowledge relevant to developing novel cyberweapons could enable global-scale harm if accessed by bad actors [54, 63]. While data filtering before pretraining could mitigate such risks, precisely auditing and filtering data at the required scale remains impractical [56, 84, 1].

Post-training methods like reinforcement learning from human feedback discourage models from using undesired capabilities on common inputs [4, 55], but leave the underlying capabilities intact. As a result, the model remains vulnerable to attacks that can elicit these capabilities, including adversarial prompts, jailbreaks, or direct finetuning [87, 38, 83]. To address this vulnerability, recent work has tried using machine unlearning to remove undesired capabilities from LLMs [53, 69, 6]. However, existing unlearning methods also suppress capabilities rather than remove them [32, 18]. The supposedly unlearned capabilities can be restored through a few steps of finetuning, leaving the fundamental challenge of achieving true capability removal unsolved [48, 49, 23, 33].

---

[*]Core contributors. Author contributions in Appendix G.

[†]Correspondence to `brucelws@seas.upenn.edu`.

[‡]We share our code implementation publicly through GitHub.

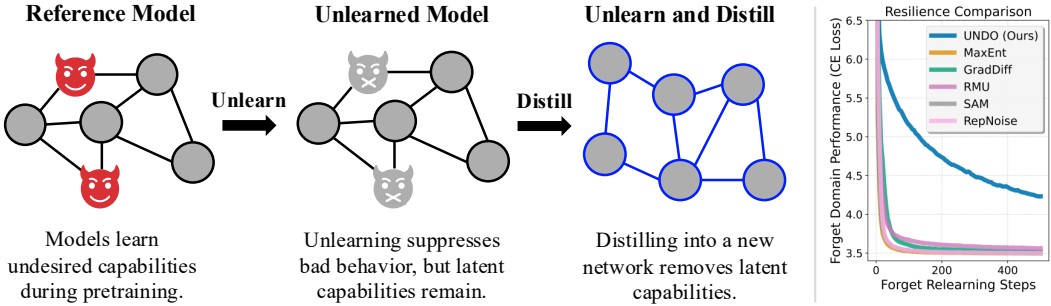

Figure 1: **Distillation robustifies unlearning**. Existing LLM unlearning methods suppress undesired behavior, but are reversible using a small amount of finetuning. We show that distilling the suppressed model into a randomly initialized network significantly increases resilience against reacquiring the undesired behavior. Our method substantially outperforms other robust unlearning baselines, including RepNoise [61] and SAM [23].

We present a simple but powerful observation: distillation robustifies unlearning. When an unlearned model is distilled into a randomly initialized student, desired behavior is transferred while undesired capabilities are left behind (Figure 1). Across diverse domains, including language, arithmetic, and weapons of mass destruction, we find that distillation consistently improves resistance to adversarial relearning attacks. Remarkably, distilled models relearn harmful capabilities at rates comparable to an oracle model [25] trained from scratch with data filtering.

This finding has immediate practical implications. Distillation is commonly used in frontier AI development [30, 86]. For example, LLM developers distill models to reduce inference costs [74, 75]. By applying unlearning methods before distillation, these developers can produce models that are strongly resistant to relearning attacks.

Building on this insight, we introduce UNDO (Unlearn-Noise-Distill-on-Outputs), which distills an unlearned model into a partially noised copy of itself, enabling a tradeoff between compute cost and unlearning robustness. With this framework, we approximate the robustness of gold-standard data filtering at a fraction of the cost, offering a new frontier in safe and scalable model deployment. We summarize our contributions as follows:

- In Section 3, we provide evidence of a limitation in behavioral training for robust unlearning in LLMs. Even when achieving perfect behavioral agreement with an unlearning oracle [25], models retain latent capabilities that can be easily restored through finetuning.

- In Section 4, we show that distillation robustifies unlearning. Training a new, randomly initialized model on the outputs of an unlearned model will transfer desired behavior while leaving latent capabilities behind.

- In Section 5, we propose UNDO, which distills an unlearned model's outputs into a noised copy of itself. This method enables a novel compute-robustness tradeoff that interpolates between mere behavioral suppression and fully robust unlearning.

- We benchmark UNDO against a variety of unlearning methods on language and arithmetic tasks, extending the Pareto frontier of retain performance versus forget unlearning robustness. We also show competitive performance and robustification of unlearning on the WMDP benchmark [42].

## 2   Robust Unlearning

Machine unlearning seeks to remove specified knowledge from a trained model while preserving the model's overall functionality. This goal has been operationalized in different ways [41]. We follow recent work concerned with removing undesired capabilities from LLMs in a way that is robust to adversarial elicitation [67, 71]. We consider the following problem:

**Robust Unlearning.** A robust unlearning method accepts the following inputs: (i) a *Reference Model*: a neural network with capabilities to be removed; (ii) a *Retain Set*: a collection of examples demonstrating desired capabilities; (iii) a *Forget Set*: a collection of examples demonstrating undesired

capabilities; (iv) *Training Data*: a collection of unlabeled examples. The method produces a new model that is evaluated according to (a) a *Retain Evaluation*: a measure of the model's desired capabilities; (b) an *Elicited Forget Evaluation*: a measure of the model's latent undesired capabilities. The goal is to produce a model with a high retain performance and low elicited forget performance.

**Elicitation Mechanism.** We use finetuning on the forget set, commonly referred to as a relearning attack, as our elicitation method. Relearning attacks have repeatedly proven to be among the most effective ways to elicit unlearned knowledge or capabilities in supposedly unlearned models [31, 33, 60]. Therefore, recent studies treat resistance to such finetuning-based probes as the primary criterion for robust unlearning [11, 23, 16]. For rigorous evaluation, we apply several relearning attacks at different learning rates and measure the maximum elicited forget performance of all attacks.

**Reference vs. Oracle Models.** In our framework, the *reference model* represents a trained model that contains both desired and undesired capabilities (e.g., a pretrained LLM). We also define an *oracle model* as one trained on the reference model's pretraining dataset with all content directly related to the undesirable capability perfectly filtered out. The oracle model often represents a gold standard solution to the robust unlearning problem, simulating perfect capability removal by never learning undesired capabilities in the first place [50, 88].

The capabilities-focused definition used above differs from machine unlearning used in privacy and fairness applications [29, 90]. Machine unlearning in privacy and fairness applications assumes the retain and forget sets partition the reference model's training data, and aims to produce an oracle model equivalent to training solely on the retain set [25]. In contrast, the capabilities-focused definition uses retain and forget sets as curated datasets reflecting desired and undesired capabilities, which need not correspond directly to literal subsets of the training data [13, 17]. Despite this conceptual shift, similar methods are often applied across both settings; however, these methods generally do not achieve robustness [78, 34].

## 3    Oracle Matching Does Not Guarantee Robust Unlearning

Unlearning methods often finetune the outputs of the reference model. For example, Gradient Ascent combined with Gradient Descent maximizes cross-entropy loss on the forget set and minimizes it on the retain set [50]. The aspiration of such procedures is oracle matching: attaining outputs indistinguishable from those of a model that never encountered the undesired data.

In this section, we demonstrate the limitations of this aspiration empirically. We finetune a reference model to match an oracle model's outputs on both the retain and forget sets to achieve effectively perfect behavioral agreement. Yet, when both models are subsequently finetuned on the forget set, the oracle-matched reference model relearns significantly faster than the oracle model. This shows that optimizing for behavioral outputs is likely insufficient for robust unlearning.

**Datasets and Models.** We conduct experiments in a language setting and an arithmetic setting, defined below. For each setting, we train a reference model with both retain and forget capabilities, and an oracle model with only retain capabilities by construction. More details on how these datasets and reference models were generated can be found in Appendix A.

- **Language.** The retain set is English documents from FineWebEdu [58]; the forget set is Korean documents from FineWeb2 [59]; the training data is the union of these two sets. The *reference model* is a 100M parameter model based on the Gemma 2 architecture (see Table 1 for details) pretrained on 2B tokens, 1B from retain data and 1B from forget; the *oracle model* is the same 100M model architecture pretrained on 1B tokens of the retain set. The retain and forget evaluations are cross-entropy loss on held-out portions of the retain and forget sets, respectively, each containing 500K tokens.

- **Arithmetic.** The retain set is addition/subtraction statements (equations and word problems) from a synthetic arithmetic dataset described in Appendix A; the forget set is multiplication and division from the same dataset; the training dataset is the union of these with (English) documents from FineWebEdu [58]; each containing 500K arithmetic questions for training. The *reference model* is trained for 1000 steps on the training data; the *oracle model* is trained for 1000 steps on the training data with the forget set replaced with additional English Fineweb data. The retain and forget evaluations measure accuracy on addition/subtraction and multiplication/division problems, using held-out portions of the retain and forget sets.

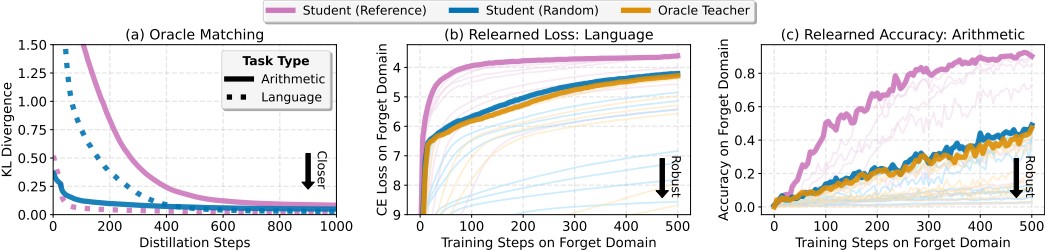

Figure 2: **Matching oracle behavior doesn't guarantee robust unlearning.** (a) KL divergence during distillation shows behavioral alignment with Oracle Teacher. (b-c) Despite this alignment, reference models matched to the oracle (**Student (*Reference*)**) exhibit rapid relearning of undesired capabilities when finetuned on the forget set, compared to the randomly initialized model matched to the oracle (**Student (*Random*)**) and the oracle itself (**Oracle Teacher**). Results highlight that an ideal unlearned behavior on the surface is insufficient for ensuring robustness against relearning.

**Oracle Matching Training.** We match the reference model's outputs with those of the oracle model using the Kullback-Leibler Divergence (KL Divergence) as the loss function. For comparison, we also match a randomly initialized model to the oracle model in the same way. The hyperparameters of the distillation setup can be found in Table 3 and the loss curves are shown in Figure 2 (a), where we use the label Student (*Reference*) or Student (*Random*) depending on whether the student in the distillation setup is the reference model or the randomly initialized model, respectively. In both settings, the distillation loss tends to zero.

**Relearning.** After distillation, we apply relearning attacks to all three models, Student (*Reference*), Student (*Random*), and Oracle Teacher, for 500 steps on data from the forget set. The hyperparameters for this training are in Table 4. We show the corresponding learning and accuracy curves for multiple learning rates in Figure 2 (b) and (c). The bolded lines correspond to the worst-case adversary (maximum elicitation across learning rates) for that number of training steps. In both the arithmetic and language settings, Student (*Reference*) quickly learns the forget distribution compared to the Student (*Random*) and Oracle Teacher models, which are more robust. This pattern holds both for the worst-case adversary and uniformly across all learning rates.

As illustrated in Figure 2, distilling the oracle's outputs into a randomly initialized student (Student (*Random*)) results in a model whose relearning speed on the forget distribution closely matches that of the oracle itself. Given that existing machine unlearning methods provide practical means of approximating oracle outputs, an important next question is whether this robustness to relearning remains after replacing true oracle outputs with approximations. In the next section, we demonstrate empirically that they do.

## 4   Distillation Robustifies Unlearning

The previous section showed that training the reference model to match oracle behavior is insufficient for robust unlearning. However, training a randomly initialized model to match the same oracle outputs does produce a robust unlearning effect. Here, we investigate whether this robustness persists when replacing the oracle with approximations from standard unlearning methods.

**Unlearning Methods.** We experiment with three unlearning methods from the literature: Gradient Difference (GradDiff) [50], Maximizing Entropy (MaxEnt) [44], and Representation Misdirection for Unlearning (RMU) [42]. These methods represent three major paradigms in LLM unlearning: gradient-based manipulation, output distribution matching, and representation-level interventions.

**Unlearning Method A: Gradient Difference (GradDiff)** applies opposing gradient updates to forget and retain data. Given a model parameterized by $\theta$, GradDiff minimizes the objective $\mathcal{L}(\theta) := -L_{\text{forget}}(\theta) + L_{\text{retain}}(\theta)$, where both loss terms are cross-entropy losses. The gradient updates aim to maximize loss on forget examples (pushing probability mass away from undesired outputs) while minimizing loss on retain examples (preserving desired model behavior).

**Unlearning Method B: Maximizing Entropy (MaxEnt)** increases uncertainty in the model's outputs on forget data while preserving performance on retain data. MaxEnt minimizes $\mathcal{L}(\theta) := L_{\text{uniform}}(\theta) + L_{\text{retain}}(\theta)$, where $L_{\text{uniform}}$ is a KL divergence term that pushes the model's output distribution toward uniformity on forget examples, and $L_{\text{retain}}$ is the standard cross-entropy loss on retain examples. Rather than directly opposing learned associations as in GradDiff, MaxEnt makes the model outputs maximally uninformative on undesired tasks.

**Unlearning Method C: Representation Misdirection for Unlearning (RMU)** operates at the representation level rather than the output level. It minimizes $\mathcal{L}(\theta) := L_{\text{misdirect}}(\theta) + \alpha \cdot L_{\text{preserve}}(\theta)$, where $L_{\text{misdirect}}$ uses mean squared error (MSE) loss to push the model's internal activations at specific layers toward a random direction for forget examples, and $L_{\text{preserve}}$ applies MSE to maintain similarity to the original model's activations for retain examples.

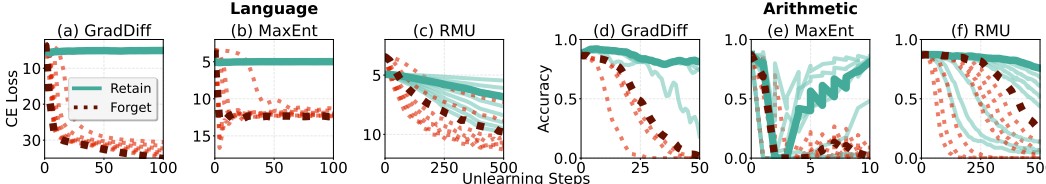

Figure 3: **Comparing unlearning methods.** (a–c) Unlearning trends across hyperparameters for our language setup, where we select configurations that maximize retain performance while minimizing forget performance for distillation (see Figures 4 and 5). (d–f) Corresponding trends in arithmetic.

**Experimental Setup.** We investigate whether distillation can enhance the robustness of unlearning methods against relearning attacks. Our experimental protocol consists of two phases: (i) applying unlearning methods (GradDiff, MaxEnt, or RMU) to a pretrained model (as shown in Figure 3), and (ii) distilling this suppressed model into a randomly initialized model of identical architecture using forward KL divergence. We refer to this method as **Unlearn-and-Distill**.

We observe relearning speed when models are subjected to relearning attacks, comparing unlearning-only models, Unlearn-and-Distill models, and oracle baselines. We defer the discussion of other relevant experimental details to Appendix B.

**Result: Distillation makes unlearning more resilient to relearning.** Figure 4 presents the relearning curves for our experiments across both language and arithmetic domains. The plots track how quickly each model reacquires the forget capability when subjected to relearning attacks. We observe that models that underwent unlearning followed by distillation (Unlearn-and-Distill) are significantly more resistant to relearning compared to unlearning-only counterparts (Unlearn). Notably, in some cases, the distilled models' relearning trajectories closely approximate those of the gold standard (Data Filtering). This robustness improvement holds whether we measure average performance across learning rates (Figure 10) or worst-case adversarial performance (Figure 4).

While RMU combined with distillation shows less impressive results in the arithmetic domain in Figure 4 (f), it still offers marked improvement over the unlearning-only model. We observe that such underperformance can occur when the initial retain-forget trade-off established by the unlearning method is less favorable. In Figure 3, RMU achieved only 62% retain and 6.8% forget accuracy, compared to MaxEnt's superior 80% and 1.3% (Appendix B.1). This suggests that distillation effectiveness can depend on the quality of the teacher model's retain-forget tradeoff.

**Why This Matters.** To contextualize the experiment, we began with custom-pretrained language models containing dual capabilities. For the language setup, approximately 50% of the pretraining data contributed to desired (retain) capabilities and 50% to undesired (forget) capabilities. Standard unlearning methods use minimal labeled data (less than 0.01% of pretraining data) to suppress undesired capabilities almost perfectly. However, this shallow suppression is vulnerable to relearning attacks. An alternative solution would be data filtering with full retraining, but this approach requires labeling all pretraining data, which is often infeasible at scale. Unlearn-and-Distill bridges this gap by transferring the suppressed model's behavior to a fresh parameter space using unlabeled pretraining data for distillation, achieving substantial robustness improvements without the extensive labeling requirements of data filtering.

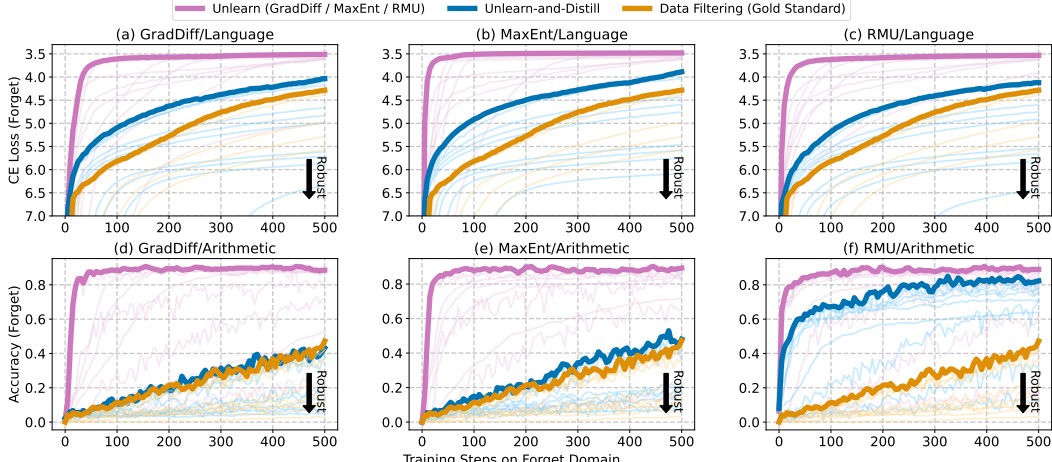

Figure 4: **Unlearn-and-Distill boosts robustness to relearning.** (a-c) Relearning trends for the language forget domain (Korean), comparing unlearning-only methods (GradDiff, MaxEnt, RMU) against models with an additional distillation step, measured against the gold standard of full retraining. We highlight the least favorable learning curve for each method. (d-f) Relearning trends for the arithmetic forget domain (Multiplication & Division).

These results provide some empirical evidence supporting claims of prior work about the challenges of robust unlearning. They have argued that quick relearning of supposedly unlearned capabilities stems from latent information preserved in the parameter space [92, 57, 66] due to properties like plasticity [47] and representational entanglement [93, 50]. Our experiments confirm that when we apply standard unlearning methods to a model, the resulting model relearns quickly under finetuning. However, when we distill this unlearned model into a randomly initialized student, the student relearns substantially slower as there was no latent capability to begin with.

**Extension to Post-Training Unlearning.** While our main experiments distill into randomly initialized networks to remove capabilities acquired during pretraining, this approach also applies when undesired behavior is learned during post-training. In such cases, the original base model (which never possessed the undesired capability) can serve as the distillation target. In Appendix F, we demonstrate this on the TOFU benchmark [50].

## 5  Trading Compute for Unlearning Robustness

So far, we have demonstrated that distilling an unlearned model into a randomly initialized model significantly enhances robustness against relearning attacks. However, this approach introduces substantial computational costs, as we are training a student model from scratch. Moving beyond the basic Unlearn-and-Distill method established in Section 4, we now investigate whether the method can be adapted to achieve different levels of robustness with different amounts of compute. That is, can we trade off robustness to reduce compute costs.

In this section, we conduct experiments that generalize the Unlearn-and-Distill approach in Section 4 with a three-step process. (i) Unlearn: create a behaviorally suppressed model (teacher) by applying standard unlearning methods. (ii) Noise: create a student model by perturbing the weights of the suppressed model, damaging it. (iii) Distill: repair this damaged student by distilling to recover the teacher's original behavior. The controlled perturbation enables us to interpolate between the original parameters and full reinitialization. We refer to this method as **Unlearn-Noise-Distill-on-Outputs**, or simply, **UNDO**. We define the perturbed model as:

$$\theta_{\text{perturbed}} = (1 - \alpha)\theta_{\text{suppressed}} + \alpha\beta N$$

where mixing coefficient $\alpha \in [0, 1]$, noise scale $\beta \in \mathbb{R}^+$, and $N$ represents sampled noise. We sample $N$ using Xavier initialization, though other random noise sampling methods could be explored. Intuitively, the term $(1 - \alpha)\theta_{\text{suppressed}}$ shrinks [2] the suppressed model's parameters, and the term

$\alpha\beta N$ injects noise. As we vary $\alpha$ from 0 to 1, we can view $\theta_{\text{perturbed}}$ as interpolating between the original paramterization and full reinitialization. We observe later in Section 6 that simply shrinking the parameters (setting $\beta = 0$) can also be effective, as the key idea is to globally damage the network to varying degrees, though including $\beta$ adds expressivity to the formulation. Note that when $\alpha = 1$ and $\beta = 1$, the perturbation reduces to random initialization.

**Experimental Setup**. We start by perturbing the models that are suppressed with MaxEnt from Section 4 and follow the same protocols for distillation using forward KL divergence on the outputs. For both language and arithmetic settings, we experiment with various values[1] of $\alpha$ while fixing $\beta = 0.1$. For each perturbation level, we perform distillation until the model reaches at least 95% of the teacher model's performance on the retain set or completes training on one full epoch of pretraining data, whichever comes first. Low-level experimental details are available in Appendix C.

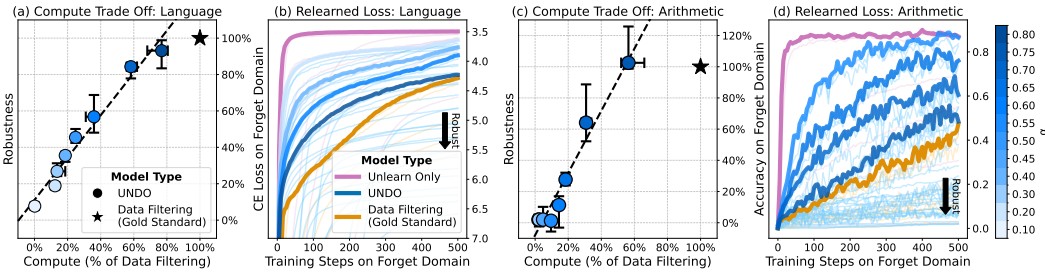

Figure 5: **Unlearning robustness scales with more perturbation.** (a, c) UNDO scaling trend for $\alpha$ between 0.1 and 0.8 and $\beta = 0.1$, showing trade-off between robustness measured as $(P_{\text{UNDO}} - P_{\text{Unlearn Only}})/(P_{\text{Data Filtering}} - P_{\text{Unlearn Only}})$ where $P$ is forget performance, and compute measured as $S_{\text{UNDO}}/S_{\text{Data Filtering}}$ where $S$ is training steps. Points denote median values, error bars show variation across five random seeds. (b) Relearning trends for Korean domain with $\alpha = \{0.2, 0.4, 0.6, 0.8\}$. (d) Relearning trends for Multiplication & Division with $\alpha = \{0.55, 0.65, 0.7, 0.75\}$.

**Result: UNDO enables trading off compute for robustness to relearning.** Figure 5 demonstrates that increasing the perturbation parameter $\alpha$ enhances robustness against relearning attacks while requiring proportionally more computation to recover the teacher model's performance on the retain set. This relationship appears roughly linear for language unlearning in Figure 5 (a) and exhibits a steeper initial curve for arithmetic unlearning in Figure 5 (c). Figures 5 (b, d) show varying degrees of resistance to relearning across different $\alpha$ values, with higher values consistently demonstrating improved robustness in both domains. For language unlearning, even relatively small perturbations ($\alpha = 0.2$) produce noticeable improvements in relearning resistance, while arithmetic unlearning shows meaningful improvements at higher perturbation levels ($\alpha \geq 0.55$).

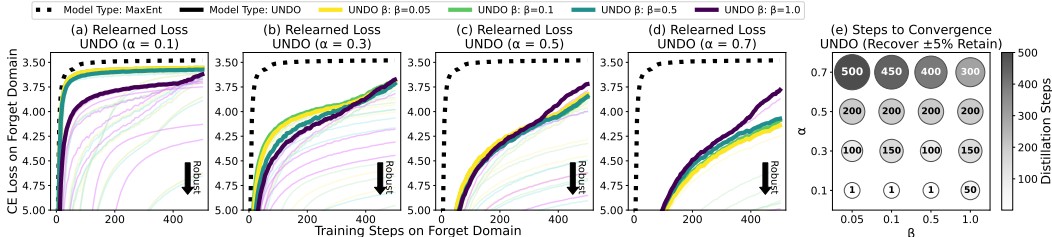

Figure 6: **Effects of varying $\alpha$ and $\beta$ noise parameters.** (a-d) Relearning curves for UNDO models with fixed mixing coefficient $\alpha = \{0.1, 0.3, 0.5, 0.7\}$ while varying noise scale $\beta = \{0.05, 0.1, 0.5, 1.0\}$. Higher mixing coefficients ($\alpha$) consistently produce more robust unlearning (flatter curves), while the effect of noise scale ($\beta$) is more nuanced. (e) Computational cost matrix showing distillation steps required to recover within 5% of the teacher's retain performance across $\alpha$ and $\beta$ combinations.

---

[1]We test $\alpha \in \{0.1, 0.2, 0.3, 0.4, 0.5, 0.55, 0.6, 0.65, 0.7, 0.75, 0.8\}$ for language and arithmetic tasks.

To learn more about the interplay between mixing coefficient ($\alpha$) and noise scale ($\beta$), we conduct a hyperparameter sweep as reported in Figure 6. Changes in mixing coefficient ($\alpha$) produce a more pronounced effect on robustness than the noise scale ($\beta$), which has a less significant impact. This pattern holds across all settings where models are distilled to within $\pm 5\%$ of the teacher's retain performance. While incorporating standard noise from Xavier initialization can help improve robustness for very low $\alpha$ values ($\alpha = 0.1$), we find that for most $\alpha$ values tested, the specific value of $\beta$ had less impact. Interestingly, we observe that using lower $\beta$ values (0.05, 0.1, or 0.5) typically worked similarly well or better than $\beta = 1.0$ in our language setup. The computational cost matrix in Figure 6 (e) confirms that $\alpha$ is the primary driver of compute requirements, with higher values often requiring more distillation steps.

## 6 Comparisons with Other Unlearning Methods

We now compare UNDO against existing robust unlearning methods to evaluate their effectiveness across two key metrics, retain performance and resistance to relearning. The ideal unlearning method would preserve high performance on desired tasks while remaining resistant to adversarial attempts to recover forget capabilities through finetuning. We first compare diverse unlearning methods in our language and arithmetic tasks in Figure 7, subjecting models to increasingly stronger adversarial relearning attacks.

We also compare against several approaches that address the robust unlearning problem using different mechanisms to resist relearning attacks. Sharpness-Aware Minimization (SAM) [23] optimizes for flat loss regions where small parameter perturbations maintain similar outputs, making the model less sensitive to finetuning. Representation Noising (RepNoise) [61] combines unlearning with noise injection at the representation level, creating interference in the model's internal representations of forget capabilities. UNDIAL [19] also uses distillation for unlearning but differs from our approach by not applying global parameter corruption techniques like noising. For UNDO, we fix $\alpha$ to 0.6 and $\beta$ to 0.1, and distill from a MaxEnt-suppressed model with varying retain thresholds[2] to obtain different points along the retain-forget trade-off curve. Figure 5 suggests that higher $\alpha$ values would yield greater robustness, while lower $\alpha$ values would yield less. We discuss other relevant experimental details in Appendix D.

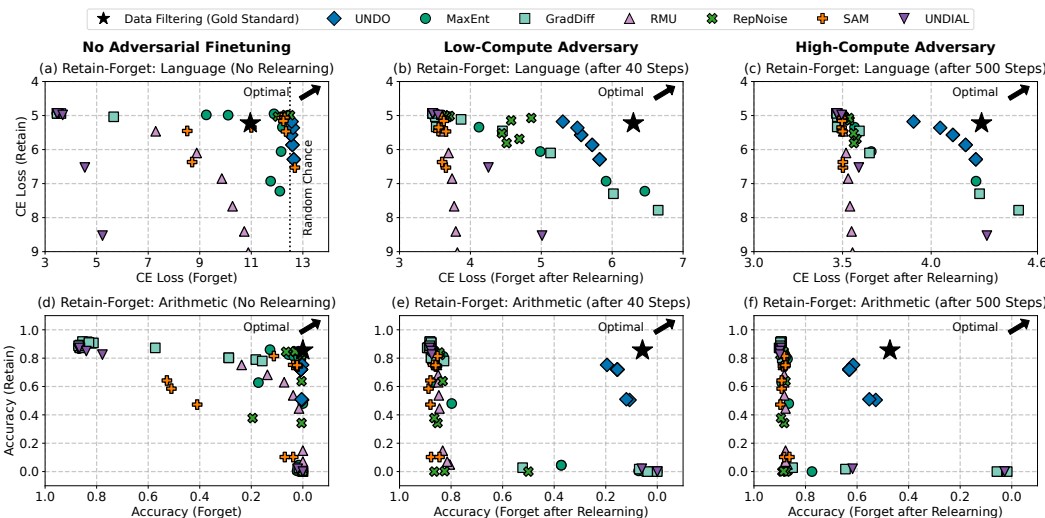

Figure 7: **Comparing unlearning methods across different adversarial strengths.** (a, d) Initial performance after unlearning but before adversarial attacks. (b, e) Relearned forget performance after moderate relearning (40 steps). (c, f) Performance after extensive relearning (500 steps).

As Figure 7 shows, while more compute-efficient unlearning methods (GradDiff, MaxEnt) achieve good initial retain-forget trade-offs before relearning attacks, they rapidly degrade under adversarial pressure. In general, we observe that methods designed for robustness (SAM, RepNoise) also show

---

[2]We test {0.05, 0.09, 0.13, 0.17, 0.21, 0.25, 0.29, 0.33} retain thresholds for language and arithmetic tasks.

significant performance deterioration when subjected to stronger relearning attacks. In contrast, UNDO with MaxEnt maintains more robust unlearning performance across all explored attack strengths. This creates a new Pareto frontier that approaches the gold standard of data filtering with full retraining but requires less compute and data labeling.

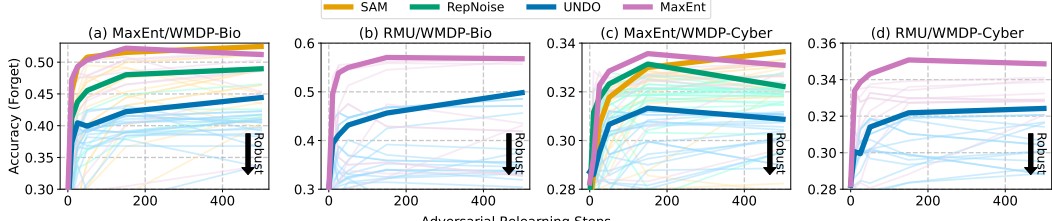

Figure 8: **UNDO makes MaxEnt and RMU more resilient to relearning WMDP.** Relearning trends averaged across 4 seeds for WMDP-Bio (a, b) and WMDP-Cyber (c, d). The retain/forget performance was measured on MMLU/the WMDP benchmark. Best adversaries are bolded.

Moving beyond our synthetic language and arithmetic unlearning tasks, we also test our methods on the more realistic Weapons of Mass Destruction Proxy (WMDP) benchmark [42]. For this setup, we use Gemma-2-2B [74] rather than custom pretrained models. Following our established methodology, we first apply standard unlearning using either MaxEnt or RMU, then apply UNDO with $\alpha = 0.25$ and $\beta = 0$. We compare this approach against the other robust unlearning baselines. We evaluate robustness against seven diverse adversarial scenarios involving different data mixtures, question-answer formats, model perturbations, and learning rate variations (See Appendix D.2).

Figure 8 shows that UNDO consistently improves relearning resilience across all WMDP configurations. These results have immediate practical implications. When model distillation is already planned for efficiency, incorporating unlearning before distillation provides robustness benefits at minimal additional cost. This approach integrates capability removal into existing distillation workflows, providing a practical path to robust unlearning.

**Limitations.** In our WMDP experiments on a pretrained Gemma-2-2B model, the UNDO model achieves 4.88% lower on MMLU on average. While this places UNDO on the Pareto frontier, other methods achieve a similar trade-off between robust forget and retain performance in this setting, as shown in Figure 11. This contrasts with the prior arithmetic and language setting results, where we see clear gains on the Pareto frontier in Figure 7. We hypothesize this difference comes from the fact that we undertrain in the WMDP setting, distilling on only 0.015% of the size of the original pretraining corpus compared to higher percentages in the other synthetic settings. We discuss this more in Appendix D.2.7. We expect that model developers would not encounter the same challenges and would have sufficient access to compute, pretraining data, and their proprietary methods to accelerate the process. Distillation is considerably more costly than finetuning-based unlearning methods, but we expect this cost to be justified when robust capability removal is critical.

## 7   Related Work

Machine unlearning aims to remove the influence of specific training examples from a trained model. This term was first introduced by Cao and Yang [10], which explored adjusting a trained model's outputs to mimic retraining without the data to be forgotten. Machine unlearning has been a significant research focus for differential privacy, implementing the "right to be forgotten" [62] through frameworks that ensure removal of specific datapoints [39, 3, 28, 46, 77]. The field has evolved into exact unlearning [8, 24, 14, 52] and approximate unlearning [27, 65, 43, 79]. Exact unlearning targets that the unlearned model's parameters exactly match those of a model retrained from scratch without the data to be forgotten [26], while approximate unlearning relaxes this constraint to require only probabilistically similar output distributions [70].

For LLMs, exact unlearning becomes impractical due to scale and non-convexity [44], leading recent studies to frame unlearning as an optimization problem that approximates the behavior of the retrained model [21, 50, 36, 45, 37, 88, 81]. LLM unlearning typically involves finetuning using objectives like maximizing prediction loss on forget sets [35, 89, 5, 22, 92] or aligning predicted

token distributions with target distributions using KL divergence [40, 82, 12, 80]. Recent studies demonstrate that existing LLM unlearning methods achieve behavioral suppression rather than true unlearning [32, 92, 76], remaining vulnerable to finetuning attacks that quickly recover suppressed knowledge [33, 49, 18]. This motivates our student-teacher approach [15] for robust unlearning.

# 8 Conclusion

There is a disconnect between what a model does and what it *could do*. Our findings illuminate this gap in two ways: first, by showing that even oracle-matched models retain latent capabilities vulnerable to adversarial elicitation; and second, by demonstrating that distillation discards these capabilities while preserving desired behavior. This insight transforms distillation, a standard practice in LLM development, into a security measure, offering a practical path to robust capability removal.

## Acknowledgments

We thank Henrik Marklund for very insightful discussions. We are grateful to Rishub Tamirisa for the guidance in navigating WMDP benchmarking procedures and to Eric Easley for sharing valuable strategies for WMDP dataset cleaning and productive discussions about potential improvements to our method. Vivek Hebbar, Andis Draguns, and Jake Mendel offered helpful comments on the abstract. We thank Iftekhar Uddin and Laura Vaughan for facilitating access to computational resources and funding support. We thank Lisa Wong for help with the design of Figure 1. We thank four anonymous reviewers for their time spent on our manuscript and for providing constructive feedback.

Bruce W. Lee's work on this project was partially supported by funding from Open Philanthropy. We also thank MATS for facilitating our collaboration and providing the institutional framework that made this research possible.

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

# Appendices

# A  Experimental Details for Section 3

In Section 3, we present oracle matching experiments. Here, we present the specific experimental details used to obtain the results presented.

**Dataset**    Our process starts from collecting datasets to facilitate our language and arithmetic unlearning experiments. For language experiments, we utilize two sources: (1) HuggingFaceFW/fineweb-2[3] Korean subset, providing non-English language examples, and (2) HuggingFaceFW/fineweb-edu[4], containing English language examples with high-quality content [59, 58]. We sample 10 million rows from each source.

For arithmetic experiments, we generate 1 million arithmetic examples covering four basic operations: addition, subtraction, multiplication, and division. We control the range of operands, using integers from 1 to 50 for addition and subtraction, and integers from 1 to 20 for multiplication and division. For division problems, we ensure clean division with no remainders by constructing problems where the dividend is a product of the divisor and quotient.

The arithmetic dataset consists of two formats:

- **Equation format:** Direct mathematical expressions (e.g., "12 + 7 = 19", "36 / 9 = 4")
- **Word problems:** Natural language scenarios generated using templates with randomized variables including names, objects, places, and quantities

Word problems are constructed using templates. For example, addition problems might involve combining collections of objects ("Emma has 8 marbles and Jack has 13 marbles. How many marbles do they have altogether?"). Division problems might involve distributing items ("There are 24 cookies in a container. If each student gets six cookies, how many students will receive cookies?").

Both language and arithmetic datasets have separate test sets. For language data, we allocated 1,000,000 tokens for validation, which consists of 500,000 tokens for the retain domain (English) and 500,000 tokens for the forget domain (Korean). For arithmetic data, we allocated 1,000,000 questions for validation, which consists of 500,000 questions for the retain domain (addition and subtraction) and 500,000 questions for the forget domain (multiplication and division).

**Data Processing**    All datasets are processed using the Google Gemma-2-2b tokenizer. Documents are chunked to a maximum length of 2,048 tokens, with documents shorter than 50 tokens being filtered out. Processing is done in parallel using multiple CPU cores to tokenize and format the data efficiently. We use a fixed random seed of 42 for data sampling and shuffling to ensure reproducibility.

**Model Architecture**    We use scaled-down versions of the Gemma-2 architecture for our language and arithmetic unlearning experiments. The model architectures and their key parameters are detailed in Table 1.

| Parameter | Gemma-2-0.1B | Gemma-2-0.3B |
|---|---|---|
| Total Parameters | ∼100M | ∼300M |
| Hidden Layers | 14 | 14 |
| Attention Heads | 8 | 8 |
| Hidden Size | 320 | 768 |
| Intermediate Size | 1280 | 3072 |
| Head Dimension | 128 | 128 |

Table 1: Model Architecture Parameters

The Gemma-2-0.1B model is used for the language unlearning experiments, while the Gemma-2-0.3B model is used for the arithmetic unlearning experiments. We intentionally vary this to show results across different model sizes. We chose these parameters based on the ratio used in the original Gemma-2 architecture to maintain the model's core properties.

---

[3]https://huggingface.co/datasets/HuggingFaceFW/fineweb-2
[4]https://huggingface.co/datasets/HuggingFaceFW/fineweb-edu

**Pretraining**   We then pretrain our Oracle and Base models. Table 2 details the hyperparameters used during pretraining. We reference these hyperparameters from [72, 91, 7].

| Hyperparameter | Language Pretraining | | Arithmetic Pretraining | |
|---|---|---|---|---|
| | Base | Oracle | Base | Oracle |
| Model Architecture | Gemma-2-0.1B | Gemma-2-0.1B | Gemma-2-0.3B | Gemma-2-0.3B |
| Parameters | 100M | 100M | 300M | 300M |
| Training Data | English + Korean | English | All Ops + English | Add/Sub + English |
| Data Mixture Ratio | 1:1 | - | 1:3 | 1:7 |
| Learning Rate | 4e-4 | 4e-4 | 4e-4 | 4e-4 |
| Min Learning Rate | 4e-5 | 4e-5 | 4e-4 | 4e-4 |
| Batch Size | 5 | 5 | 15 | 15 |
| Gradient Accum. Steps | 80 | 80 | 24 | 24 |
| Weight Decay | 0.1 | 0.1 | 0 | 0 |
| Epochs | 1 | 1 | 1 | 1 |
| Maximum Steps | - | - | 1000 | 1000 |
| LR Schedule | Cosine | Cosine | Cosine | Cosine |
| Warmup Steps | 50 | 50 | 50 | 50 |
| Max Sequence Length | 2048 | 2048 | 256 | 256 |

Table 2: Pretraining Hyperparameters

For the *language* models, the Base model is trained on both English and Korean data with equal probabilities, while the Oracle model is trained exclusively on English data. Both models used identical optimization parameters. For the *arithmetic* models, the Base model is trained on all four arithmetic operations (addition, subtraction, multiplication, division) mixed with English text at a 1:3 ratio. The Oracle model is trained only on addition and subtraction operations mixed with English at a 1:7 ratio. We end up using $\sim 1B$ pretraining tokens for Gemma-2-0.1B (Language, Oracle), $\sim 2B$ tokens for Gemma-2-0.1B (Language, Base), $\sim 100M$ tokens for Gemma-2-0.3B (Arithmetic, Oracle), $\sim 100M$ tokens for Gemma-2-0.3B (Arithmetic, Base).

All models were trained using the AdamW optimizer with cosine learning rate scheduling. We used a shorter sequence length (256 tokens) for arithmetic models compared to language models (2048 tokens) due to the inherently shorter nature of arithmetic problems.

**Oracle Matching**   We implemented oracle matching using knowledge distillation with forward KL divergence loss [30] on the pretraining dataset, as also explored in Gemma 2 technical report [74]. As illustrated in Figure 2(a), we examined two configurations: (1) distillation from an Oracle teacher to a randomly initialized model and (2) distillation from an Oracle teacher to a pretrained Base model. Table 3 presents the hyperparameters for these distillation procedures.

| Hyperparameter | Language Distillation | Arithmetic Distillation |
|---|---|---|
| Teacher Model | Gemma-2-0.1B (Oracle) | Gemma-2-0.3B (Oracle) |
| Student Model | Gemma-2-0.1B (Base or Rand Init) | Gemma-2-0.3B (Base or Rand Init) |
| Distillation Data | English + Korean | All Ops + English |
| Data Mixture Ratio | 1:1 | 1:3 |
| Learning Rate | 4e-4 | 7e-4 |
| Min Learning Rate | 4e-5 | 7e-5 |
| Batch Size | 4 | 15 |
| Gradient Accum. Steps | 160 | 24 |
| Weight Decay | 0.1 | 0 |
| Maximum Steps | - | 1000 |
| Epochs | 1 | 1 |
| LR Schedule | Cosine | Cosine |
| Warmup Steps | 50 | 50 |
| Max Sequence Length | 2048 | 256 |

Table 3: Oracle Matching Hyperparameters

**Relearning** To evaluate the robustness of unlearning methods, we conduct relearning experiments to assess whether unlearned capabilities could be easily recovered. Notably, finetuning is often considered a strong form of latent capability elicitation, so it can provide insights into whether knowledge is truly removed from the model or just suppressed during oracle matching. For all relearning experiments, we use the final student model from the oracle matching procedure as the starting point. Training proceeds for 500 steps, with validation performed every five steps to track both retention and recovery of capabilities. Table 4 presents the hyperparameters used for our relearning experiments.

We assume a significantly stronger adversary compared to contemporary works in LLM unlearning, and show that our methods introduced in Sections 4 and 5 are effective across weaker and stronger adversaries. Our Language Relearning experiments train ∼8M tokens, and Arithmetic Relearning experiments train ∼2M tokens. On top of this, we do an adversarial learning rate search for all relearning runs over {1e-3, 7e-4, 4e-4, 1e-4, 7e-5, 5e-5, 1e-5, 7e-6, 4e-6, 1e-6}. In all relearning experiments (Figure 2, Figure 4, Figure 5), we always mark the adversarially optimal performance (the best performance the adversary could obtain by varying the learning rate) at each validation step in thicker lines.

| Hyperparameter | Language Relearning | Arithmetic Relearning |
|---|---|---|
| Relearning Data | Korean | Mul/Div + English |
| Data Mixture Ratio | - | 1:3 |
| Batch Size | 8 | 15 |
| Gradient Accumulation Steps | 1 | 1 |
| Maximum Steps | 500 | 500 |
| Warmup Steps | 1 | 1 |
| Learning Rate | [1e-3, 7e-4, 4e-4, 1e-4, 7e-5, 5e-5, 1e-5, 7e-6, 4e-6, 1e-6] | |
| Evaluation Steps | 5 | 5 |
| Max Sequence Length | 2048 | 256 |

Table 4: Relearning Hyperparameters

# B    Experimental Details for Section 4

In Section 4, we report unlearn and distill experiments. Many experimental details are similar to what was reported in Appendix A.

**Dataset / Data Processing / Model Architecture / Pretraining / Relearning**    See Appendix A.

**Unlearning Methods**    In Figure 3, we report unlearning trends for three methods: GradDiff, MaxEnt, and RMU. To perform a hyperparameter search, we first manually search for the key hyperparameters (see Table 5).

| Hyperparameter | Language Unlearning | | | Arithmetic Unlearning | | |
|---|---|---|---|---|---|---|
| | **GradDiff** | **MaxEnt** | **RMU** | **GradDiff** | **MaxEnt** | **RMU** |
| Forget Data | Korean | Korean | Korean | Mult/Div | Mult/Div | Mult/Div |
| Retain Data | English | English | English | Add/Sub | Add/Sub | Add/Sub |
| Batch Size | 4 | 4 | 4 | 40 | 40 | 40 |
| Max Steps | 100 | 100 | 500 | 10 | 10 | 500 |
| Max Length | 2048 | 2048 | 2048 | 256 | 256 | 256 |
| Alpha | - | - | 1200 | - | - | 200 |
| RMU Layers | - | - | 5–11 | - | - | 5–11 |
| End Layer | - | - | 11 | - | - | 11 |
| c | - | - | 6.5 | - | - | 6 |

Table 5: Unlearning Hyperparameters

### B.1 Unlearning Learning Rate Sweeps

We do learning rate sweeps across eight learning rates per method. We choose the one that gives the best trade-off between the retain and forget performance. The chosen values (marked with thick lines in Figure 3) are marked in bold below.

**GradDiff (Language, CE Loss)**

1e-5 Initial Retain: 4.9359 → Final Retain: 5.0264 ‖ Initial Forget: 3.4668 → Final Forget: 30.5592

2e-5 Initial Retain: 4.9359 → Final Retain: 4.9924 ‖ Initial Forget: 3.4668 → Final Forget: 31.5783

3e-5 Initial Retain: 4.9359 → Final Retain: 4.9967 ‖ Initial Forget: 3.4668 → Final Forget: 32.3159

4e-5 Initial Retain: 4.9359 → Final Retain: 5.0085 ‖ Initial Forget: 3.4668 → Final Forget: 32.9097

5e-5 Initial Retain: 4.9359 → Final Retain: 5.0216 ‖ Initial Forget: 3.4668 → Final Forget: 33.5308

6e-5 Initial Retain: 4.9359 → Final Retain: 5.0334 ‖ Initial Forget: 3.4668 → Final Forget: 34.1291

7e-5 Initial Retain: 4.9359 → Final Retain: 5.0563 ‖ Initial Forget: 3.4668 → Final Forget: 34.685

**8e-5 Initial Retain: 4.9359 → Final Retain: 5.0762 ‖ Initial Forget: 3.4668 → Final Forget: 35.129**

**GradDiff (Arithmetic, Accuracy)**

6e-6 Initial Retain: 0.875 → Final Retain: 0.9125 ‖ Initial Forget: 0.8675 → Final Forget: 0.8375

7e-6 Initial Retain: 0.875 → Final Retain: 0.9075 ‖ Initial Forget: 0.8675 → Final Forget: 0.8325

**8e-6 Initial Retain: 0.875 → Final Retain: 0.815 ‖ Initial Forget: 0.8675 → Final Forget: 0.015**

9e-6 Initial Retain: 0.875 → Final Retain: 0.74 ‖ Initial Forget: 0.8675 → Final Forget: 0

1e-5 Initial Retain: 0.875 → Final Retain: 0.7475 ‖ Initial Forget: 0.8675 → Final Forget: 0

2e-5 Initial Retain: 0.875 → Final Retain: 0.1825 ‖ Initial Forget: 0.8675 → Final Forget: 0

3e-5 Initial Retain: 0.875 → Final Retain: 0.8025 ‖ Initial Forget: 0.8675 → Final Forget: 0.28

4e-5 Initial Retain: 0.875 → Final Retain: 0.79 ‖ Initial Forget: 0.8675 → Final Forget: 0.1775

**MaxEnt (Language, CE Loss)**

1e-5 Initial Retain: 4.9359 → Final Retain: 4.9439 ‖ Initial Forget: 3.4672 → Final Forget: 11.3281

2e-5 Initial Retain: 4.9359 → Final Retain: 4.9397 ‖ Initial Forget: 3.4672 → Final Forget: 11.6977

3e-5 Initial Retain: 4.9359 → Final Retain: 4.942 ‖ Initial Forget: 3.4672 → Final Forget: 11.9844

4e-5 Initial Retain: 4.9359 → Final Retain: 4.9466 ‖ Initial Forget: 3.4672 → Final Forget: 12.1715

5e-5 Initial Retain: 4.9359 → Final Retain: 4.9551 ‖ Initial Forget: 3.4672 → Final Forget: 12.2584

6e-5 Initial Retain: 4.9359 → Final Retain: 4.9633 ‖ Initial Forget: 3.4672 → Final Forget: 12.3114

7e-5 Initial Retain: 4.9359 → Final Retain: 4.976 ‖ Initial Forget: 3.4672 → Final Forget: 12.3608

**8e-5 Initial Retain: 4.9359 → Final Retain: 4.9895 ‖ Initial Forget: 3.4672 → Final Forget: 12.3808**

**MaxEnt (Arithmetic, Accuracy)**

6e-5 Initial Retain: 0.875 → Final Retain: 0.86 ‖ Initial Forget: 0.8675 → Final Forget: 0.1275

7e-5 Initial Retain: 0.875 → Final Retain: 0.83 ‖ Initial Forget: 0.8675 → Final Forget: 0.0525

8e-5 Initial Retain: 0.875 → Final Retain: 0.8075 ‖ Initial Forget: 0.8675 → Final Forget: 0.0175

**9e-5 Initial Retain: 0.875 → Final Retain: 0.795 ‖ Initial Forget: 0.8675 → Final Forget: 0.0125**

1e-4 Initial Retain: 0.87 → Final Retain: 0.81 ‖ Initial Forget: 0.87 → Final Forget: 0.035

2e-4 Initial Retain: 0.87 → Final Retain: 0.4825 ‖ Initial Forget: 0.87 → Final Forget: 0

3e-4 Initial Retain: 0.87 → Final Retain: 0.045 ‖ Initial Forget: 0.87 → Final Forget: 0.015

4e-4  Initial Retain: 0.87 → Final Retain: 0.005 ‖ Initial Forget: 0.87 → Final Forget: 0.0225

**RMU (Language, CE Loss)**

1e-5  Initial Retain: 4.9359 → Final Retain: 5.4614 ‖ Initial Forget: 3.4668 → Final Forget: 7.2717

2e-5  Initial Retain: 4.9359 → Final Retain: 6.0976 ‖ Initial Forget: 3.4668 → Final Forget: 8.8457

**3e-5  Initial Retain: 4.9359 → Final Retain: 6.8464 ‖ Initial Forget: 3.4668 → Final Forget: 9.8125**

4e-5  Initial Retain: 4.9359 → Final Retain: 7.6585 ‖ Initial Forget: 3.4668 → Final Forget: 10.2192

5e-5  Initial Retain: 4.9359 → Final Retain: 8.4008 ‖ Initial Forget: 3.4668 → Final Forget: 10.6792

6e-5  Initial Retain: 4.9359 → Final Retain: 9.0205 ‖ Initial Forget: 3.4668 → Final Forget: 10.8073

7e-5  Initial Retain: 4.9359 → Final Retain: 9.4909 ‖ Initial Forget: 3.4668 → Final Forget: 10.958

8e-5  Initial Retain: 4.9359 → Final Retain: 9.8514 ‖ Initial Forget: 3.4668 → Final Forget: 11.2771

**RMU (Arithmetic, Accuracy)**

6e-6  Initial Retain: 0.875 → Final Retain: 0.7575 ‖ Initial Forget: 0.8675 → Final Forget: 0.2275

7e-6  Initial Retain: 0.875 → Final Retain: 0.68 ‖ Initial Forget: 0.8675 → Final Forget: 0.1375

**8e-6  Initial Retain: 0.875 → Final Retain: 0.615 ‖ Initial Forget: 0.8675 → Final Forget: 0.0675**

9e-6  Initial Retain: 0.875 → Final Retain: 0.52 ‖ Initial Forget: 0.8675 → Final Forget: 0.035

1e-5  Initial Retain: 0.875 → Final Retain: 0.4425 ‖ Initial Forget: 0.8675 → Final Forget: 0.025

2e-5  Initial Retain: 0.875 → Final Retain: 0.14 ‖ Initial Forget: 0.8675 → Final Forget: 0

3e-5  Initial Retain: 0.875 → Final Retain: 0.065 ‖ Initial Forget: 0.8675 → Final Forget: 0

4e-5  Initial Retain: 0.875 → Final Retain: 0.05 ‖ Initial Forget: 0.8675 → Final Forget: 0

## B.2  Distillation

**Distillation**    In Figure 4, we report relearning trends for the unlearned, distilled, and gold standard. We use the same dataset that was used to pretrain respective models as distillation data. We also ensure that the distilled model undergoes the same number of update steps as its pretrained counterpart. We perform distillation with the hyperparameters in Table 6. Additionally, we report the distillation trend in Figure 9. To find information about Data Filtering (Gold Standard) models, refer to Appendix A.

| Parameter | Language Distillation | | | Arithmetic Distillation | | |
|---|---|---|---|---|---|---|
| | **GradDiff** | **MaxEnt** | **RMU** | **GradDiff** | **MaxEnt** | **RMU** |
| Teacher Model | Unlearn Only[5] | Unlearn Only | Unlearn Only | Unlearn Only | Unlearn Only | Unlearn Only |
| Student Model | Random Init | Random Init | Random Init | Random Init | Random Init | Random Init |
| Distillation Data | Kor + Eng | Kor + Eng | Kor + Eng | All Ops + Eng | All Ops + Eng | All Ops + Eng |
| Interleave Probs | [0.5, 0.5] | [0.5, 0.5] | [0.5, 0.5] | [0.75, 0.25] | [0.75, 0.25] | [0.75, 0.25] |
| Batch Size | 4 | 4 | 4 | 15 | 15 | 15 |
| Grad. Accum. Steps | 60 | 60 | 60 | 24 | 24 | 24 |
| Epochs | 1 | 1 | 1 | 1 | 1 | 1 |
| Learning Rate | 9e-4 | 9e-4 | 9e-4 | 7e-4 | 7e-4 | 7e-4 |
| Min Learning Rate | 7e-4 | 7e-4 | 7e-4 | 7e-5 | 7e-5 | 7e-5 |
| Max Steps | - | - | - | 1000 | 1000 | 1000 |
| Warmup Steps | 50 | 50 | 50 | 50 | 50 | 50 |
| Scheduler | Cosine | Cosine | Cosine | Cosine | Cosine | Cosine |
| Weight Decay | 0.1 | 0.1 | 0.1 | 0.0 | 0.0 | 0.0 |
| Max Seq. Length | 2048 | 2048 | 2048 | 256 | 256 | 256 |

Table 6: Distillation Hyperparameters

---

[5]This refers to the GradDiff/MaxEnt/RMU unlearned models, which are also marked as "Unlearn Only" in Figure 4.

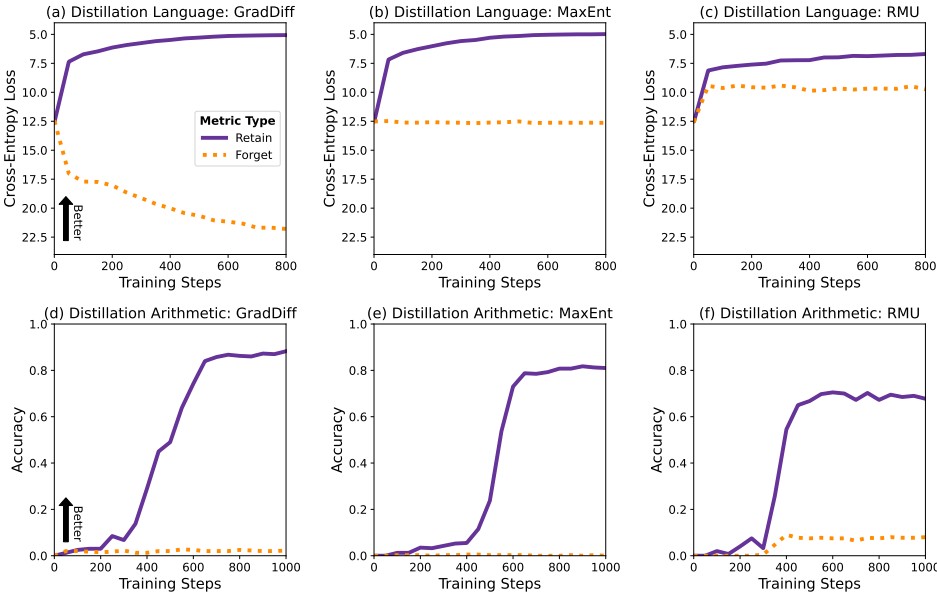

Figure 9: Distillation Trends

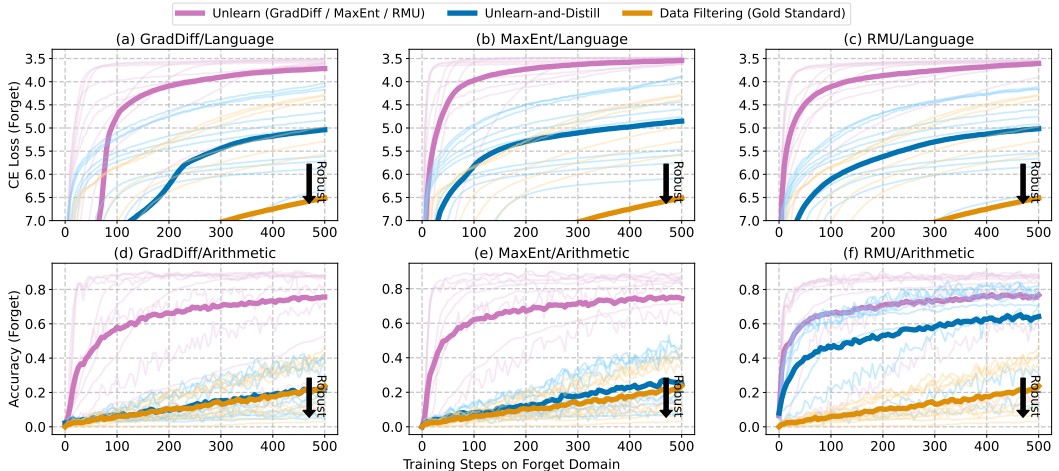

Figure 10: Figure 4 shown with average lines bolded.

## C Experimental Details for Section 5

In Section 5, we introduce UNDO as an approach to leverage random noising as an alternative to full random initialization.

**Dataset / Data Processing / Model Architecture / Pretraining / Relearning** See Appendix A.

**Unlearning Method** We choose the best MaxEnt method from Appendix B based on its favorable trade-off between retain and forget performance. For language experiments, we use a learning rate of 8e-5, which demonstrated the highest trade-off score while maintaining strong performance on both domains (Initial Retain: 4.9359 → Final Retain: 4.9895, Initial Forget: 3.4672 → Final Forget: 12.3808). For arithmetic experiments, we use a learning rate of 9e-5, which similarly achieved strong performance (Initial Retain: 0.875 → Final Retain: 0.795, Initial Forget: 0.8675 → Final Forget: 0.0125).

**Noising Method** The novel element introduced in Section 5 is controlled parameter corruption, allowing us to interpolate between the original model and random noise. This provides a more flexible approach compared to binary choices of keeping original parameters or fully reinitializing them. The algorithm applies a weighted combination of the original parameters and scaled random noise based on two hyperparameters: $\alpha$ controls the proportion of corruption applied, while $\beta$ controls the magnitude of the noise. The pseudocode is as follows:

```python
def do_corruption(model, noise_alpha, noise_beta=0.1, seed=42):
    # Loop through all parameters and add random noise
    assert 0 <= noise_alpha <= 1
    assert 0 <= noise_beta

    for param in model.parameters():
        if param.requires_grad:
            # Initialize corruption tensor
            corruption = torch.zeros_like(param.data)

            # Generate appropriate noise based on parameter dimensionality
            if len(param.data.shape) == 2:
                # For weight matrices (2D tensors), use Xavier init
                noise = torch.nn.init.xavier_uniform_(
                    torch.empty_like(param.data)
                )
            elif len(param.data.shape) == 1:
                # For bias vectors (1D tensors), use zeros
                noise = torch.zeros_like(param.data)
            else:
                raise RuntimeError(
                    f"Unsupported parameter shape: {param.data.shape}"
                )

            # Scale the noise by beta
            corruption = noise_beta * noise

            # Apply weighted combination
            param.data=(1 - noise_alpha)*param.data+ noise_alpha*corruption

    # Move model to appropriate device
    model.to(device)
```

For our experiments, we systematically vary $\alpha$ from 0 to 0.8, while maintaining $\beta$ at a constant value of 0.1 in Figure 5. This allows us to evaluate the effect of gradually increasing levels of parameter corruption on both retention of desired capabilities and forgetting of undesired ones.

**Distillation Method** See Appendix B. We follow the same distillation protocol established in our previous experiments, using the noised models (obtained after applying the corruption function) as the student models in the distillation process.

## D    Experimental Details for Section 6

In Section 6, we compare UNDO against several other unlearning methods to evaluate their effectiveness in maintaining a balance between retain performance and resistance to relearning. This appendix provides the detailed experimental setup and hyperparameters used for these comparison experiments.

### D.1    Language and Arithmetic Unlearning

**Dataset / Data Processing / Model Architecture / Pretraining** We use the same datasets, processing methods, model architectures, and pretraining procedures as described in Appendix A.

Specifically, we conduct experiments on both the language (English/Korean) task using Gemma-2-0.1B and the arithmetic (addition-subtraction/multiplication-division) task using Gemma-2-0.3B.

**Baseline Unlearning Methods**    We compare several unlearning approaches:

- **GradDiff**: Gradient Difference based unlearning with the forget domain.
- **MaxEnt**: Maximum Entropy based unlearning that maximizes the model's uncertainty on the forget dataset while maintaining performance on the retain dataset.
- **RMU**: Representation Misdirection for Unlearning, which modifies internal model representations for selected layers.
- **SAM**: Sharpness-Aware Minimization that optimizes for flat loss regions to resist relearning.
- **RepNoise**: Representation Noising that combines standard unlearning with noise injection at the representation level.
- **UNDIAL**: Unlearning via distillation approach without global model damaging.

**Learning Rate Sweeps**    For GradDiff, MaxEnt, RMU, and UNDIAL, we conducted extensive learning rate sweeps to find optimal configurations. We tested logarithmically spaced learning rates: 3e-3, 8e-3, 3e-4, 8e-4, 3e-5, 8e-5, 3e-6, 8e-6, 3e-7, 8e-7, 3e-8, and 8e-8. For each method, we chose the learning rate that provided the best trade-off between retain performance and forget degradation.

**RepNoise Configurations**    For RepNoise experiments, we used the best MaxEnt hyperparameters as the base configuration and then conducted a grid search over the following RepNoise-specific parameters:

| Alpha ($\alpha$) | Beta ($\beta$) |
| --- | --- |
| 0.1 | 0.0001 |
| 0.1 | 0.001 |
| 0.5 | 0.01 |
| 0.5 | 0.1 |
| 1.0 | 0.1 |
| 1.0 | 1.0 |
| 2.0 | 2.0 |
| 4.0 | 4.0 |

Table 7: RepNoise parameter configurations, where alpha controls the strength of the noise loss and beta controls the ascent loss.

**SAM Configurations**    For SAM experiments, we also started with the best MaxEnt hyperparameters and performed a parameter sweep over the SAM perturbation radius ($\rho$) parameter:

| Rho ($\rho$) |
| --- |
| 0.001 |
| 0.003 |
| 0.005 |
| 0.01 |
| 0.02 |
| 0.04 |
| 0.07 |
| 0.1 |

Table 8: SAM perturbation radius values explored. The value 0.01 is recommended by the original SAM paper.

**UNDO Configuration**    For our proposed UNDO method, we fixed the corruption parameters at $\alpha = 0.6$ and $\beta = 0.1$ based on our findings in Section 5. We used the MaxEnt as the unlearning algorithm before applying random noising and distillation. We also experimented with various retain

threshold values {0.05, 0.09, 0.13, 0.17, 0.21, 0.25, 0.29, 0.33} to obtain different points along the retain-forget trade-off curve.

**Common Training Parameters**  All language experiments used the following common parameters unless otherwise specified for a particular method:

- Batch size: 4

- Max sequence length: 2048

- Scheduler: Cosine

- Weight decay: 0.0

- Gradient clipping threshold: 1.0

For arithmetic experiments, we used:

- Batch size: 40 (GradDiff: 20, RepNoise/SAM variants: 5 with gradient accumulation steps of 8)

- Max sequence length: 256

- Scheduler: Cosine

- Weight decay: 0.0

- Gradient clipping threshold: 1.0

**Method-Specific Parameters**  Each unlearning method required specific hyperparameters:

- **GradDiff**: Alpha (retain-forget tradeoff) = 1 for language, Alpha (retain-forget tradeoff) = 15 for arithmetic

- **RMU**: Layers = [5-11], End layer = 11, Alpha = 1200, c = 6.5 for language; Alpha = 200, c = 6 for arithmetic

**Relearning Evaluation**  To evaluate robustness against relearning, we fine-tuned each unlearned model on the forget dataset (Korean or multiplication/division) for varying numbers of steps (0, 40, and 500) to simulate increasingly strong adversarial attacks. We monitored both the retain performance (on English or addition/subtraction) and forget performance to assess how well each method maintained the desired knowledge while suppressing the forgotten capabilities.

## D.2   WMDP

We apply our method to the WMDP [42] benchmark on both the Cyber and Bio domain and compare to existing unlearning methods. We use Gemma-2-2b [73] as the base model.

### D.2.1   Evaluation/Benchmark

For the retain evaluation we use MMLU, evaluating the model with 5-shot on a subset of 40% of the full MMLU evaluation and report accuracy. For the forget evaluation we use WMDP Bio/Cyber depending on the domain use the entire evaluation, using zero-shot prompting.

### D.2.2   Datasets

WMDP includes WMDP forget and retain corpora for each domain. However, it is common practice to use other datasets as well. For example the original WMDP paper introduces RMU [42] using Wikitext [51] as the retain dataset.

Applying unlearning methods using the original forget and retain datasets as well as public auxiliary datasets such as Wikitext proves difficult, especially for the biology domain and methods other than RMU. We hypothesize this is due the datasets possessing numerous differences that could be used to differentiate them. This allows the unlearning methods to achieve low loss on their objectives but fail to generalize to the evaluation. To remedy this, we create a dataset that eliminates most undesired differences and emphasizes the relevant differences in the domains. We extract a question answer dataset from each corpora, Bio retain, Bio forget, Cyber retain, and Cyber forget, by prompting

Gemini flash to write questions and answer from text sampled from the corpora. Additionally, we extract questions and answer from English Wikipedia using the same process. The following is an example of a question and answer generated for the Cyber forget dataset.

> **Question:** How does Address Space Layout Randomization (ASLR) complicate shellcode injection?
> **Answer:** Address Space Layout Randomization (ASLR) randomizes the memory addresses of key program areas, making it harder for attackers to predict where shellcode should be placed in memory for successful exploitation.

We experimented with using different datasets and combinations of datasets to optimize the tradeoff between forgetting and maintaining retain performance. Ultimately, we use the forget question-answer dataset for the forget set and a 50/50 mix of the retain question-answer dataset and Wikitext for the retain set.

### D.2.3 Baseline Unlearning Methods

We use two baseline unlearning methods, RMU and MaxEnt.

Both RMU and MaxEnt involve optimizing a loss that is the sum of a retain component and a forget component. For WMDP, we parameterize this combination with $\alpha$ such that $loss = \alpha * retain\_component + (1 - \alpha) * forget\_component$ where $0 < \alpha < 1$.

For MaxEnt, we use a variation in which the loss maximizes entropy on the forget set while minimizing KL divergence between the base model and unlearned model on the retain set. For RMU, we use the same loss as language/arithmetic settings. We use the parameters in Table 9.

| Parameter | Value |
|---|---|
| **General Training Parameters** | |
| Batch size | 40 |
| Sequence length | 256 |
| Steps | 90 |
| Learning rate | Fixed (no scheduler) |
| Weight decay | 0 |
| Warmup steps | 0 |
| **RMU-Specific Parameters** | |
| $c$ value | 80 |
| RMU layers | $\{10, 11, 12, 13, 14, 15\}$ |
| RMU end layer | 15 |

Table 9: Hyperparameter Settings

We run hyperparameter sweeps on alpha and learning rate for both methods, then chose three points across the frontier to use in the experiment. We run four seeds of each method and apply adversaries to each to measure elicited forget performance. Of the three points we use the one with the highest retain performance as the teacher for the UNDO models. Ultimately, the models in Table 10 are chosen.

### D.2.4 Distillation/UNDO

We apply UNDO using an auxiliary dataset. The dataset consists of the following datasets, specified by the Huggingface name, and mixed at the percentages shown in Table 11.

### D.2.5 Relearning Adversaries

We extensively explore relearning adversaries via small batches of relearning runs. To ensure fairness and avoid only selecting adversaries that do well on baseline methods, we always run configurations on the UNDO method. We run the most promising configurations across all seeds of the methods, and we run all configurations on UNDO. We report the metrics from the best adversary for each method.

| Domain | Method | Configuration | Retain Performance Ranking |
|--------|--------|---------------|----------------------------|
| Bio | RMU | $\alpha = 0.5, \text{lr} = 5 \times 10^{-5}$
$\alpha = 0.3, \text{lr} = 1 \times 10^{-4}$
$\alpha = 0.1, \text{lr} = 1 \times 10^{-4}$ | Best
Medium
Worst |
| | MaxEnt | $\alpha = 0.3, \text{lr} = 2 \times 10^{-5}$
$\alpha = 0.1, \text{lr} = 2 \times 10^{-5}$
$\alpha = 0.75, \text{lr} = 5 \times 10^{-5}$ | Best
Medium
Worst |
| Cyber | RMU | $\alpha = 0.5, \text{lr} = 2 \times 10^{-5}$
$\alpha = 0.2, \text{lr} = 2 \times 10^{-5}$
$\alpha = 0.1, \text{lr} = 2 \times 10^{-5}$ | Best
Medium
Worst |
| | MaxEnt | $\alpha = 0.2, \text{lr} = 2 \times 10^{-5}$
$\alpha = 0.3, \text{lr} = 3.5 \times 10^{-5}$
$\alpha = 0.3, \text{lr} = 5 \times 10^{-5}$ | Best
Medium
Worst |

Table 10: Selected Models Ordered by Performance Trade-off

| Dataset | Proportion |
|---------|------------|
| HuggingFaceFW/fineweb-edu (`subset_name="sample-10BT"`) | 0.35 |
| legacy-datasets/wikipedia (`subset_name="20220301.en"`) | 0.35 |
| Magpie-Align/Magpie-Llama-3.3-Pro-1M-v0.1 | 0.05 |
| Magpie-Align/Magpie-Llama-3.1-Pro-1M-v0.1 | 0.04 |
| Magpie-Align/Llama-3-Magpie-Pro-1M-v0.1 | 0.04 |
| Magpie-Align/Magpie-Gemma2-Pro-534K-v0.1 | 0.04 |
| Magpie-Align/Magpie-Phi3-Pro-1M-v0.1 | 0.04 |
| Magpie-Align/Magpie-Qwen2.5-Pro-1M-v0.1 | 0.04 |
| Magpie-Align/Magpie-Qwen2-Pro-1M-v0.1 | 0.04 |
| Wikipedia question-answer | 0.01 |
| **Total** | 1.00 |

Table 11: Auxiliary Dataset Composition

We explore seven configurations and vary the learning rate for each.

We have a variety of techniques and datasets that we vary to form 6 adversary configurations.

**Dataset Descriptions:**

1. **forget/retain:** Consists of a 50/50 mixed forget and retain WMDP corpora from the corresponding domain.

2. **forget/retain-qa:** Consists of a 50/50 mixed forget and retain question-answer dataset from the corresponding domain.

3. **wiki-qa:** Consists of a 50/25/25 mix of wikitext, forget question-answer, and retain question-answer from the corresponding domain.

**Sequence Length Descriptions:**

- **Standard:** Sequence length 256
- **Long:** Sequence length 1024

**Technique:**

- **Shrink-perturb:** When applied, we apply shrink perturb to the weights as formulated in the UNDO method with $\alpha = 0.05$ before relearning.

We test diverse adversaries, as each existing method may be selectively robust against certain attacks. For example, RepNoise was much more susceptible to the shrink-perturb relearning, while being

| Adversary Name | Dataset | Sequence Length | Shrink-perturb |
|---|---|---|---|
| forget/retain | 1 | standard | no |
| forget/retain-qa | 2 | standard | no |
| forget/retain-long | 1 | long | no |
| forget/retain-qa-long | 2 | long | no |
| wiki-qa-long | 3 | long | no |
| sp-forget/retain | 1 | standard | yes |
| sp-forget/retain-qa | 2 | standard | yes |
| sp-bio-wiki-qa-long | 3 | long | yes |

Table 12: Configuration of the seven adversary techniques used to evaluate unlearning robustness. Dataset values correspond to: (1) original forget/retain data, (2) question-answer formatted data, and (3) Wikipedia data with QA formatting. Sequence length indicates whether a standard or extended context was used. Shrink-perturb indicates whether the model parameters were perturbed before relearning.

more robust to other adversaries. Similarly, for the TAR method, the authors report robustness across most adversaries, but select adversaries were completely able to recover performance [71]. We were unable to find hyperparameters that maintained reasonable retain performance within our training budget.

### D.2.6 Baseline Robust Methods

We explore two robust baselines, SAM and RepNoise, in the Arithmetic and Language settings. For SAM, we set $\rho$ to 0.01, and for RepNoise we use $\beta$=1.0 and $\alpha$=1.0. Both of these methods operate in addition to MaxEnt, so we use the top two selected MaxEnt hyperparameters.

### D.2.7 WMDP Pareto Frontier and Discussion

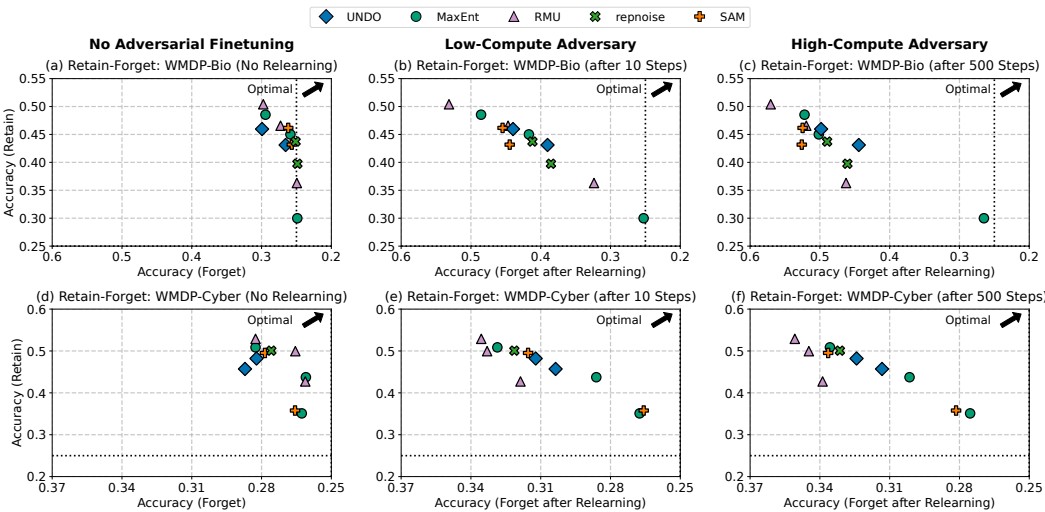

Figure 11: We measure and plot forget and retain accuracy on WMDP Bio (col 1) and Cyber (col 2), immediately after unlearning (row 1), after a low compute adversarial retraining (row two), and after a high compute adversarial retraining (row 3).

We apply MaxEnt, RMU, UNDO, RepNoise, and SAM to Gemma-2-2b. We observe UNDO is on the Pareto frontier for WMDP-Bio, and competitive with other methods for WMDP-Cyber, despite decreases in retain performance. The results are shown in Figure 11. We observe that the UNDO method has a similar robust-forget vs retain tradeoff to other baseline methods.

This contrasts with the earlier results for the language/arithmetic settings seen in Figure 7, with UNDO far exceeding the Pareto frontier set by the baseline methods. We hypothesize that this is due

to under-training the model due to limited compute and not having access to the original pretraining corpus.

More thoroughly, the key difference is in the arithmetic/language pareto Figure 7 we use $\alpha = 0.6$, whereas in WMDP pareto Figure 11, we use $\alpha = 0.25$, implying more noise in the language/arithmetic settings than WMDP. Higher alpha damage to the model's capabilities, making them harder to retrain. We show this in Figure 5, where higher alpha values are more robust and achieve lower retrained forget performance. However, when we apply higher alpha values to Gemma-2-2b, we find that it doesn't recover retain performance (see Table 13).

In Figure 5, we observe that higher alpha values require more data to recover performance. For $\alpha = 0.6$, the language and arithmetic settings require around 35% and 20% of their pretraining corpus, respectively. For WMDP, our reference model, Gemma-2-2b was originally trained on 2 trillion tokens [73], but we distill on only 300 million, 0.015% of the Gemma-2-2b pretraining corpus. However, for AI companies aiming to apply our method to their models, we expect that they have sufficient compute resources to train on a larger fraction of the pretraining corpus. Additionally, it's possible that the original pretraining corpus has better coverage or diversity to effectively learn the capabilities compared to our distillation data.

We expect that the challenges we face would not apply to the companies that could apply our methods to their models because they would have sufficient compute resources, access to the full pretraining corpora, and could even have proprietary distillation methods that allow faster distillation or better generalization compared to our setup.

| Alpha | Retrained MMLU Value |
|---|---|
| Original (0) | 0.55 |
| 0.25 | 0.49 |
| 0.35 | 0.37 |
| 0.6 | 0.25 |

Table 13: Alpha noise values applied to Gemma-2-2b and corresponding retrained MMLU values. $\beta = 0$ for all experiments.

# E    Compute requirements

We run all experiments on servers with multiple H200 or A100 GPUs. Language and arithmetic pretraining took 4xH200/1xA100 GPUs for two to three days. All distillation and UNDO runs for language are run on 4xH200 GPUs, while relearning took one H200 GPU per setup. All distillation and UNDO runs for arithmetic are run on a single A100 GPU, while relearning took one A100 GPU per setup. Our most computationally expensive experiments are WMDP UNDO runs, which take around 7 hours on one H200. Additionally, computing evaluations while relearning takes 8 minutes to do a sparse evaluation relearning run (which collects evaluations at steps 10, 25, 50, 150, and 500).

# F    Post-Review Reflection: An Alternative Setup to Study Unlearn-and-Distill

We present supplementary experiments conducted during the review period that extend our findings to additional benchmarks and attack vectors. The main concerns raised centered on the generalizability of our findings beyond simplified settings and the computational requirements of training models from scratch. We address these specifically in Appendices F.1 and F.2.

We first discuss an alternative experimental setup that enables studying the unlearn-and-distill method without reinitializing models from scratch. In this section, consider a pretrained model $M$ and a dataset $D$ that we can guarantee $M$ has never encountered during pretraining. Here, we assume the not encountering the dataset $D$ can guarantee that the pretrained model $M$ doesn't learn the capabilities associated with it.

Given the setup above, we can then finetune $M$ on $D$ to create $M'$, apply unlearning methods on $M'$ to suppress $D$, then distill $M'$ back onto the original model $M$. Since $M$ never possessed latent knowledge of $D$, it serves as an effective substitute for a randomly initialized network with respect to this specific knowledge.

Our observation throughout the paper was that once a network has learned certain capabilities during pretraining, reversing this learning requires a significant corruption of the network parameters (also see concurrent literature [85] that discusses the level of corruption needed for robust unlearning to be more likely). Indeed, the reference (unlearned) models in Figure 2 and Figure 4 demonstrated rapid relearning, and we broadly refer to the neural network properties that enable this rapid recovery as "latent traces." However, if we have data $D$ that was definitively absent from the pretraining corpus, then $M$ contains no such latent traces for $D$. This allows us to emulate the unlearn-and-distill setup from Section 4 without access to the original pretraining data of $M$.

There are challenges to obtaining an ideal $D$. Language models are essentially large token classifiers that develop circuits to use all tokens in their vocabulary, including those that would appear in $D$. Even if the semantic content of $D$ is novel, the model has seen and predicted its constituent tokens. Therefore, it's likely that keyword or semantic matching cannot guarantee the absence of latent traces.

Fortunately, this challenge is not new, and we have an established approximation. The TOFU benchmark [50] contains 200 fictional author profiles with question-answer pairs, constructed to be absent from existing corpora. Here, the specific associations between authors and their biographical details are novel. Therefore, we conduct supplementary experiments in Appendices F.1 and F.2 with the assumption that TOFU provides sufficient separation to study the distillation phenomenon in a controlled setting where $M$ effectively starts at random initialization with respect to the forget knowledge.

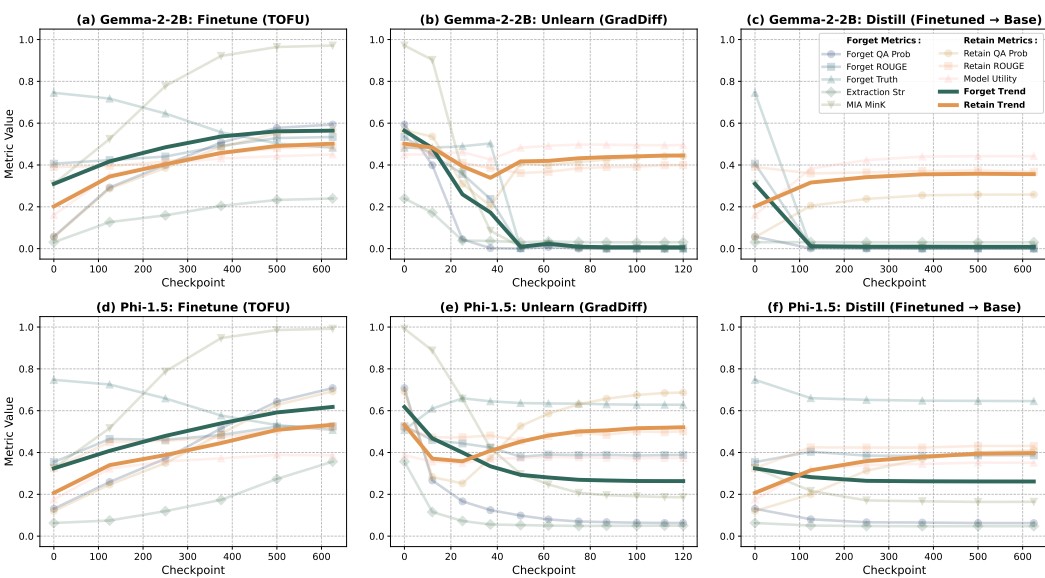

Figure 12: Training dynamics across finetuning, unlearning, and distillation phases on TOFU. Panels (a-c) show Gemma-2-2B, and panels (d-f) show Phi-1.5.

## F.1 TOFU Experiments

Following the framework presented above, we conducted experiments using the three-step protocol.

- First, we finetuned Gemma-2-2B / Phi-1.5 on the full TOFU dataset, achieving high accuracy on the fictional author questions.

- Second, we applied GradDiff (gradient ascent on the forget set; gradient descent on the retain set), suppressing the model's ability to answer questions about these authors while maintaining performance on the retain set.

- Third, we distilled the unlearned model back onto a fresh Gemma-2-2B / Phi-1.5 base model. We use the full TOFU dataset that was used to finetune the base model, including both forget and retain domains.

Figure 12 shows the training dynamics across all three phases. We conduct the experiment using the open-unlearning framework [20] and track seven commonly used unlearning metrics. During finetuning (a, d), both models learn the fictional author information, with forget metrics increasing steadily while MIA (membership inference attack) [68] scores rise to near-perfect levels. The unlearning phase (b, e) shows suppression of forget knowledge, with Forget QA probability dropping to near zero within 50 steps while retain performance stabilizes. The distillation phase (c, f) demonstrates selective knowledge transfer. The student models maintain forget metrics similar to the base model that hasn't seen TOFU, despite being trained on outputs from the full dataset, while retain metrics gradually recover to match the teacher's performance.

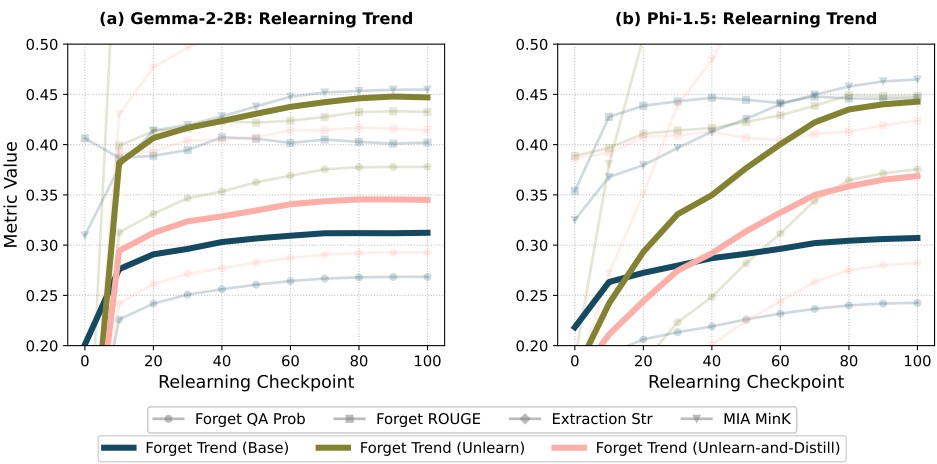

Figure 13: Relearning dynamics on TOFU forget set for (a) Gemma-2-2B and (b) Phi-1.5. Models that underwent unlearning followed by distillation show relearning rates closer to control baselines that never saw the forget authors, while unlearned-only models rapidly recover the suppressed knowledge. Individual metrics are shown in light colors with average trends in bold.

To evaluate robustness, we subjected three models to relearning attacks: the unlearned-only model, the unlearned-and-distilled model, and a control baseline (the original base model that never saw TOFU). Figure 13 shows the relearning dynamics over 100 gradient updates. The unlearned-only models rapidly recover the suppressed knowledge, approaching the performance levels seen after initial finetuning. In contrast, the unlearned-and-distilled models show substantially slower relearning, with trajectories closer to the control baselines that never possessed the forget knowledge.

These results demonstrate that the distillation robustification phenomenon holds even when using existing models as fresh networks, provided the forget knowledge was absent from their original training. The consistency across both Gemma-2-2B and Phi-1.5 architectures suggests this approach generalizes across different model families.

## F.2 Robustness Beyond Relearning Attacks

Prior work established that gradient-based finetuning represents the strongest elicitation technique for recovering unlearned capabilities [31, 49]. Nevertheless, we tested robustness against alternative attack vectors during the review period. Zhang et al. [92] demonstrated that quantization can bring back unlearned knowledge by exploiting numerical instabilities in model weights. We evaluated INT4 quantization attacks using the open-unlearning framework on our TOFU-trained models.

| Method | Precision | Forget Q&A Prob | Forget Truth Ratio | Model Utility |
|---|---|---|---|---|
| GradDiff | FP16 | 0.000 | 0.000 | 0.495 |
| GradDiff | INT4 | 0.008 | 0.185 | 0.242 |
| GradDiff + Distilled | FP16 | 0.000 | 0.001 | 0.442 |
| GradDiff + Distilled | INT4 | 0.003 | 0.017 | 0.391 |

Table 14: Quantization attack results on TOFU (Gemma-2-2B). Lower forget metrics indicate better unlearning.

Under INT4 quantization, standard unlearning shows substantial degradation of unlearning performance, with forget truth ratio jumping from 0.000 to 0.185. The distilled models demonstrate better robustness, with forget truth ratio reaching only 0.017. The distilled models also maintain better utility under quantization (0.391 vs 0.242).

These quantization results suggest that distillation creates fundamentally different internal representations rather than merely suppressing outputs. The student network, having never possessed the forget knowledge in its parameter space, lacks the latent structures that quantization exploits in the unlearned-only models. This aligns with our broader hypothesis that distillation acts as a capability filter, allowing only the knowledge actively demonstrated by the teacher to transfer to the student.

# G Author Contribution

All authors contributed to the early development of the ideas of the paper.

**Bruce W. Lee** was a core contributor to our experimental codebase and was responsible for framing and conducting most experiments reported in the paper, except WMDP. This includes Unlearn-and-Distill and UNDO experiments reported in Figures 1, 2, 3, 4, 5, 6, 7. BL implemented baselines and plotted all figures in the paper. BL wrote the majority of the later sections and later revised the earlier sections.

**Addie Foote** conducted all WMDP experiments in Figures 8, 11 and initially designed our arithmetic unlearning experiments. AF was a core contributor to our experimental codebase, making unlearning and distillation scripts, efficiency optimizations, and experiment infrastructure. AF wrote key sections on the WMDP experimental setup and the limitations discussion. AF consistently addressed the most challenging and uncertain problems, operating in the face of uncertainty, particularly with the WMDP benchmark.

**Alex Infanger** contributed significantly to the writing of the paper, the development of the problem statement, the interpretation and the discussion of experimental results, and the implementation of a pilot example of the oracle matching result. AI embarked on a semantic odyssey to improve our understanding of the robust unlearning problem.

**Leni Shor** conducted exploratory research to mechanistically validate our oracle matching experiments, helping the team gain confidence in our ideas. LS also explored formalisms that improved our understanding of the problem we were trying to solve. LS conducted initial MNIST experiments with AI, streamlining the MNIST codebase.

**Harish Kamath** achieved the first working implementation of our experiments on WMDP, which informed our later experimental approach. HK implemented and conducted experiments with noising schedulers as part of the initial UNDO framework development, and explored activation-based distillation instead of logit-based distillation.

**Jacob Goldman-Wetzler** developed the initial proof-of-concept demonstrating that distillation robustifies unlearning and reported the first positive results. JGW assisted with WMDP experiments and optimized RMU on Gemma models. JGW also contributed to activation-based distillation experiments and provided debugging support for the codebase when needed.

**Bryce Woodworth** served as research manager, organizing and facilitating team research meetings and coordinating project logistics. BW provided essential support in maintaining team communication and collaboration throughout the research process. BW contributed to copy-editing and offered valuable feedback on presentation and clarity.

**Alex Cloud** suggested combining distillation with unlearning and investigating oracle matching, and also co-led the research team. AC worked closely with the team, offering helpful input on implementation details, experimental design, and conceptual framing. He also contributed to writing.

**Alexander Matt Turner** co-led the research team and was instrumental in the initial conceptualization of the project. AT contributed to earlier iterations of the distillation approach, originally explored in the context of Gradient Routing based on his suggestions. Throughout the project, AT provided valuable feedback on experimental direction, helped analyze results, and contributed to manuscript development.

