# OpenReview forum: "Distillation Robustifies Unlearning"
_NeurIPS.cc/2025/Conference — NeurIPS 2025 spotlight_

### Official Review · Reviewer_nuYc · 2025-07-01

**Clarity:** 4
**Significance:** 3
**Originality:** 4
**Rating:** 5
**Confidence:** 4

**Summary:**

The paper tackles the problem that most machine-unlearning techniques for large LMs only suppress dangerous capabilities that are to be unlearned, leaving them latent and easy to re-learn with light fine-tuning. The methods presented in the paper are motivated by a simple observation: if you first apply any standard unlearning method to suppress the unwanted behavior and then distill that model into a randomly initialized student, the harmful capability is largely stripped away while the desired skills are retained. They demonstrate this across two unlearning tasks, arithmetic and the WMDP weapons benchmarks, showing that their Unlearn-Noise-Distill-on-Outputs (UNDO) procedure (which adds controlled parameter noising to trade compute for robustness -- i.e., taking a weighted average between a randomly initialized and the pretrained model) closes much of the gap to an expensive "retrain-from-scratch with perfect data filtering" oracle (training only on the "clean" subset of the data). UNDO consistently sits on the pareto frontier of the utility-unlearning curve, with distilled models relearning harmful tasks no faster than the oracle while preserving most of the original utility.

**Questions:**

In Fig7a, how come many methods do better than the filtering gold standard?

**Ethical Concerns:**

["NO or VERY MINOR ethics concerns only"]

**Final Justification:**

I am satisfied with the authors response. They answer my concerns, particularly these related to the limited evaluation in the original submission. I hope the new results will be incorporated into the final version of this work.

**Limitations:**

yes

**Quality:**

3

**Strengths And Weaknesses:**

The paper is clearly written. The objective is well defined, the problem which is the focus of this paper is interesting, and the presented method is motivated by experiments that show very clear differences in unlearning rate between randomly initialized and pretrained models.

Most of the experiments are carried out in relatively simplified settings, where the forget set is either an entire language or the distinction between two mathematical operators (addition and multiplication). In such a case, the distinction between the examples to be unlearned and the retain set is very significant, as they differ in domain. This contrasts with usual unlearning work that focuses on very specific concepts. Fig 8 reports results on WMDP, but does not present the utility-unlearning curve. What would it look like in this case?

I am not sure the forget-utiltiy comparison in Fig7 is completely fair. If I understand correctly, the baselines just experience unlearning, while UNDO experiences unlearning+retraining on the retain set. It can be the case that this retraining is responsible for better retain-set accuracy. Can it be the case that it’s mainly this retraining, and not the noise, that explains the better tradeoff? Did you try to ablate the noising step?

Robustness is measured via gradient-based relearning attacks (finetuning on the forget set). It would be beneficial to also compare against jailbreaking attacks in inference time.

---

> ### Author Rebuttal · Authors · 2025-07-31
>
> We are glad that the reviewer found our writing clear and findings interesting!
>
> ---
>
> ### W1: Experiments in simplified settings
>
> *From review text: “Most of the experiments are carried out in relatively simplified settings, where the forget set is either an entire language or the distinction between two mathematical operators (addition and multiplication).”*
>
> We understand this concern about domain separation. To directly address it, we tested our method on TOFU [1], a benchmark specifically designed to evaluate unlearning of specific concepts like in typical unlearning work. TOFU contains 200 synthetic author profiles with 20 Q&A pairs each. These fictional authors test unlearning where forget and retain sets are closely related.
>
> **Table: TOFU Relearning Attacks (Gemma-2-2B)**
>
> | Model | Initial Performance | Performance After 100-step Finetuning
> |-------|-------------------|----------------------------------
> | **Unlearned Only** (GradDiff) | Forget Q&A: 0.000, MIA: 0.000 | Forget Q&A: 0.378, MIA: 0.822
> | **Unlearned+Distilled** (Ours) | Forget Q&A: 0.000, MIA: 0.011 | Forget Q&A: 0.293, MIA: 0.551
> | **Control** (Never saw forget authors) | Forget Q&A: 0.057, MIA: 0.310 | Forget Q&A: 0.269, MIA: 0.455
>
> *Forget Q&A: Probability of correctly answering forget set author questions (lower is better)*
>
> *MIA: Membership Inference Attack success rate (lower is better)*
>
> Our distilled models achieve similar relearning rates to the control baseline (a model that never saw the forget authors), while unlearned-only models rapidly recover the suppressed knowledge. This demonstrates our method also works when unlearning specific concepts. That said, as we explain in our introduction, our primary motivation differs from typical unlearning work. We focus on removing broad capabilities, rather than specific facts or concepts. Our method works for both use cases.
>
> ---
>
> ### W2: WMDP utility results
>
> *From review text: "Fig 8 reports results on WMDP, but does not present the utility-unlearning curve. What would it look like in this case?"*
>
> You're right that Figure 8 didn’t include complete utility-unlearning curves. We provide these results in Figure 11 in the appendix:
>
> **Table: WMDP-Bio**
>
> | Method | MMLU | WMDP Forget |  |  |
> |--------|-------------|------------------------|------|------|
> | | | Initial | 40 steps | 500 steps |
> | MaxEnt | 0.485 | 0.294 | 0.485 | 0.522 |
> | MaxEnt+UNDO | 0.460 | 0.299 | 0.440 | 0.498 |
>
> We didn't highlight them in the main paper because interpreting WMDP results is challenging. Gemma-2-2B's pretraining likely already contains weapons and cybersecurity information, making it impossible to establish a true "never seen" baseline like we can with TOFU's fictional authors. For clarity, we also evaluated on TOFU, where we have guaranteed control over exposure:
>
> **Table: TOFU Utility-Unlearning Tradeoff**
>
> | Method | Model Utility | Forget Q&A (Initial) | Forget Q&A (50 steps)
> |--------|--------------|---------------------|----------------------
> | Unlearned Only | 0.444 | 0.000 | 0.362
> | Unlearned+Distilled | 0.442 | 0.000 | 0.283
> | Control (Gold Standard) | - | 0.057 | 0.261
>
> *Model Utility is the harmonic mean of 9 metrics measuring performance on retain set, real authors, and world facts (3 metrics each: Q&A probability, ROUGE score, and truth ratio)*
>
> Our method achieves 23% better robustness than unlearning alone while maintaining competitive utility (0.442 vs 0.444 for unlearning only). The distilled model's relearning trajectory closely matches the control, suggesting that the post-unlearning distillation step does promote a deeper knowledge removal while retaining the model's utility. This pattern holds regardless of which unlearning method (GradDiff, NPO, etc.) is used before distillation, confirming the generality of our approach.
>
> ---
>
> ### W3: Fair comparison concerns
>
> *From review text: "I am not sure the forget-utility comparison in Fig7 is completely fair. If I understand correctly, the baselines just experience unlearning, while UNDO experiences unlearning+retraining on the retain set."*
>
> You raise an important point, but we believe there may be a slight misunderstanding about how UNDO works. With UNDO, you first select any unlearning method (GradDiff, MaxEnt, etc.) with your desired retain-forget balance (Figure 3). Then, separately, you decide how much compute to invest in distillation to improve robustness and gain back the retain performance. The retain-forget trade-off is bounded by the teacher (unlearning method of choice), while robustness scales with compute (trade compute for robustness, Figure 5).
>
> Crucially, UNDO does not retrain on a labeled retain set. During distillation, we distill on the full pretraining dataset, including both retain and forget data. While the distillation does require more compute than prior unlearning methods, this step enables increased robustness. This is analogous to heavily damaging and then rewriting parts of the model internal while having a teacher who can only properly demonstrate certain capabilities. What the teacher pretends not to know, the student won't learn at all.
>
> For example, the table below shows our distillation trends where the datasets contained 50% forget data; the student only recovers capabilities the teacher demonstrates:
>
> | Language Task | Initial → Final | Arithmetic Task | Initial → Final |
> |---------------|-----------------|-----------------|-----------------|
> | English CE (Retain) | 12.52 → 4.98 | Addition Acc | 0% → 80% |
> | Korean CE (Forget) | 12.52 → 12.64 | Multiplication Acc | 0% → 0% |
>
> The student's Korean loss remains at random initialization levels (12.64) despite seeing Korean text in every batch. Similarly, multiplication accuracy stays at 0% even though 25% of the arithmetic distillation data contains multiplication problems. The initially unlearned teacher provides meaningful gradients only for retain capabilities. This property creates a unique space between existing approaches:
>
> | Method | Low Compute | High Robustness | Minimal Data Labeling |
> |--------|-------------|-----------------|----------------------|
> | Data Filtering | X | O | X (All pretraining corpus) |
> | LLM Unlearning | O | X | O (Smaller labeled set) |
> | **UNDO (Ours)** | **Tunable** | **Tunable** | **O (Same as LLM unlearning)** |
>
> Regarding Figure 7 that you specifically mentioned, all UNDO points use the same noise parameters (α=0.6, β=0.1). The different points along the retain-forget tradeoff curve come from varying when we stop distillation based on retain performance thresholds {±0.05, ±0.09, ±0.13, ±0.17, ±0.21, ±0.25, ±0.29, ±0.33}, not from different amounts of noising.
>
> ---
>
> ### W4: Robustness beyond gradient-based attacks
>
> *From review text: "Robustness is measured via gradient-based relearning attacks (finetuning on the forget set). It would be beneficial to also compare against jailbreaking attacks in inference time."*
>
> We first note that prior literature established that finetuning is the strongest elicitation technique. [2, 3] found that while fine-tuning consistently performed best across all tested setups, other attack vectors showed less consistent success. This universal vulnerability to fine-tuning, combined with its ease of implementation and replicability across research groups, motivated our focus on fine-tuning.
>
> We also tested (inference-time) quantization attacks. Recent work [5] showed that quantization can bring back unlearned knowledge:
>
> **Table: INT4 Quantization Robustness (TOFU, Gemma-2-2B)**
>
> | Method | Forget Q&A Prob | Forget Truth Ratio | Model Utility |
> |--------|----------------|-------------------|---------------|
> | GradDiff (FP16) | 0.000 | 0.000 | 0.495 |
> | GradDiff (INT4) | 0.008 | 0.185 | 0.242 |
> | GradDiff+Distilled (FP16) | 0.000 | 0.001 | 0.442 |
> | GradDiff+Distilled (INT4) | 0.003 | 0.017 | 0.391 |
>
> Our distilled models show dramatically better resilience to quantization attacks across all metrics. We highlight forget truth ratio (true utterance about the forget set authors) improvement (0.185 vs 0.017). This suggests that distillation provides defense against orthogonal attack vectors beyond just gradient-based relearning.
>
> Additionally, we have tested prompt-based and activation steering elicitation methods on our bilingual models. However, it was challenging to get clear signals, potentially due to the relatively small model sizes we used (100M-300M parameters) or limitations in applying these techniques to our specific setups.
>
> ---
>
> ### Q: Figure 7a performance
>
> This phenomenon actually has a clear explanation. Our Korean dataset (FineWeb split) contains some English characters, so training on English alone still slightly improves Korean performance, though it remains close to random (CE loss ~11-12).
>
> Methods like MaxEnt explicitly maximize entropy on the forget set, pushing outputs toward uniform distributions. This can achieve a cross-entropy loss closer to the "random" baseline than what the gold standard data filtering naturally achieves. As shown in Figure 7a, MaxEnt pushes Korean CE loss close to around 12.5, while the gold standard (trained only on English) settles around 11.
>
> A more disruptive method like GradDiff, which performs a negative gradient update against the forget set, can push Korean CE loss to the regimes of 22~25, which also explains the distillation dynamics in Figure 9. However, CE loss values above ~11 represent essentially non-linguistic outputs. The model isn't producing "better forgetting" but rather completely uninformative tokens.
>
> ---
>
> — We thank the reviewer for helpful questions! —
>
>
> [1] TOFU: A Task of Fictitious Unlearning for LLMs. COLM 24
>
> [2] WMDP Benchmark: Measuring and Reducing Malicious Use With Unlearning. ICML 24
>
> [3] The elicitation game: Evaluating capability elicitation techniques. ICML 25
>
> [4] Eight methods to evaluate robust unlearning in llms. Preprint
>
> [5] Catastrophic Failure of LLM Unlearning via Quantization. ICLR 25

---

> > ### Comment · Reviewer_nuYc · 2025-08-01
> > **Response**
> >
> > Thank you for the detailed response. I am satisfied with the answers and raise my score accordingly.

---

> > > ### Author Response · Authors · 2025-08-03
> > > **Response to the reviewer**
> > >
> > > Thank you for your thorough engagement with our work and for recognizing the contributions. We appreciate your constructive feedback throughout this process.

---

### Official Review · Reviewer_TRF9 · 2025-07-01

**Clarity:** 4
**Significance:** 3
**Originality:** 3
**Rating:** 5
**Confidence:** 4

**Summary:**

This paper addresses the critical shortcoming of existing machine unlearning methods for large language models, which often only achieve "behavioral suppression" rather than true removal of undesired capabilities. The authors demonstrate that these suppressed capabilities can be easily recovered through adversarial finetuning. The core contribution is the finding that distilling an unlearned "teacher" model into a new, randomly initialized "student" model can robustify the unlearning process, effectively transferring desired behaviors while discarding the latent, undesired capabilities. Building on this insight, the paper introduces Unlearn-Noise-Distill-on-Outputs (UNDO), a practical method that creates a tunable trade-off between unlearning robustness and computational cost by partially noising the teacher model's weights before distillation. Through extensive experiments on language modeling, arithmetic reasoning, and the Weapons of Mass Destruction Proxy (WMDP) benchmark, the authors show that their approach significantly improves resilience to relearning attacks compared to existing methods, establishing a new Pareto frontier for the trade-off between model performance and unlearning robustness.

**Questions:**

Do the authors have preliminary results for a wider variety of model families and larger model sizes? It would be interesting to see if robustness degrades with scaling.

WMDP is still a proxy. Have you tried latent knowledge probes rather than behavioral scores?

The performance drop on the MMLU retain task for the WMDP experiments is a key practical hurdle. Do you have other ideas on how to mitigate this issue without needing the original pretraining data?

The paper argues that UNDO is more efficient than the "gold standard" of data filtering with full retraining. Could you provide a more concrete, quantitative comparison of the computational cost (e.g., total FLOPs) between UNDO and full retraining for one of the smaller-scale experiments, such as the language task? This would help readers better contextualize the claimed cost savings.

The proposed UNDO method introduces noise into the weight space of the teacher model before distillation. Have you considered or experimented with alternative methods of "damaging" the teacher to prevent the transfer of latent information?

The effect of perturbation levels (alpha) seem to vary based on task (e.g. arithmetic vs language). Do you have an intuition for why this is the case? Does this suggest that the optimal alpha is highly task-dependent, and can you offer any guidance for how a practitioner might select a suitable value for a new task without performing an expensive hyperparameter sweep?

Some references are broken (showing as ??) - please fix these.

**Ethical Concerns:**

["NO or VERY MINOR ethics concerns only"]

**Final Justification:**

I have carefully read the author rebuttal and found it helpful in clarifying certain aspects of the submission. While the response addressed some of my concerns, it did not significantly alter my overall evaluation of the paper. As a result, I have decided to retain my original score.

**Limitations:**

Yes. The authors have adequately addressed the limitations of their work in a dedicated paragraph and in the appendix. They are upfront about the significant computational cost of distillation and the challenge of maintaining retain performance when the original pretraining data is unavailable. The discussion is fair and provides important context for the results.

**Paper Formatting Concerns:**

Some references are broken (showing as ??).

**Quality:**

4

**Strengths And Weaknesses:**

Strengths:

The paper tackles a highly significant and timely problem in LLM safety. The distinction between merely suppressing behavior and genuinely removing a capability is a central challenge for deploying safe models. This work offers a practical and seemingly effective method to bridge this gap, which is of great interest to the community.

The experimental evaluation is exceptionally thorough and well-designed. The authors construct a compelling narrative, starting by demonstrating the fundamental limitations of output-matching for robust unlearning. They then systematically validate their core hypothesis - that distillation robustifies unlearning - across multiple unlearning techniques and tasks. The proposed UNDO method is analyzed in detail, clearly illustrating the compute-robustness trade-off. The comparison against strong, relevant baselines under increasingly powerful relearning attacks is convincing and highlights the superiority of the proposed approach. Finally, validating the method on a realistic benchmark like WMDP strengthens the claims. The appendices are remarkably detailed, ensuring high potential for reproducibility.

The paper is very well-written, clearly structured, and easy to follow. The central idea is simple yet powerful, and it is communicated effectively through concise prose and informative figures. The logical flow from problem identification to solution proposal and validation makes the paper a pleasure to read.

While its constituent parts (unlearning, distillation) are not new, the core insight - that distillation can act as a selective filter to remove latent capabilities from an unlearned teacher - is novel and impactful. The generalization of this idea into the UNDO framework, which offers a practical knob to tune the trade-off between cost and robustness, is an original and useful contribution. The initial experiment showing that oracle-matching is insufficient for robustness is a valuable finding that helps clarify the nature of the problem for the field.

The paper contains clear ablation studies and multiple experiments supporting the main hypotheses and also answering potential questions from the readers. For instance, the plots showing effect of noise levels on robustness was very useful.

The paper uses well known and recent benchmarks to measure effects of the method (e.g. MMLU for retain performance and WMDP for forget performance).

Code will be released on a public GitHub repository, which is clearly stated in the main body of the paper. The paper also lists detailed hyperparameter settings. Both of these enhance reproducibility of this work and its usefulness to the community.

If the basic observation holds at larger scales, it offers a practical path for open-weight release: distillation is already part of most deployment pipelines, so adding “unlearn then distill” could become standard practice. That is a meaningful contribution to LLM safety.

Weaknesses:

The primary weakness, which the authors rightly acknowledge, is the high computational cost of distillation. While the paper frames this as a manageable trade-off and notes that distillation is already part of many development pipelines, the cost may still be a significant barrier to adoption, especially for organizations without access to massive compute resources or for applications where frequent unlearning is required.

A key practical limitation is the observed drop in performance on the retain task when applying UNDO to a public model without access to its original pretraining data. This currently limits the "off-the-shelf" applicability of the method and suggests that its effectiveness is still somewhat dependent on having access to high-quality, representative data for the distillation phase.

The paper provides strong empirical evidence that distillation robustifies unlearning but offers a less detailed explanation for why this is the case. The intuition that desired behavior is transferred while latent capabilities are "left behind" is compelling but high-level. A deeper analysis - perhaps investigating the information bottleneck properties of distillation in this context or examining changes in the model's internal representations - could further strengthen the paper's contribution, though this may be extensive enough for future work.

The selection of models and model sizes is very limited.

---

> ### Author Rebuttal · Authors · 2025-07-27
>
> We thank the reviewer for the thoughtful and thorough review! We first address the key concerns raised by the reviewer and then proceed to discuss other questions.
>
> ---
>
> ### W1: Access to pretraining data, off-the-shelf applicability, breadth of models tested
>
> For the threat models we consider (for example, bad actors finetuning open-weight models), mitigation becomes significantly more difficult once the weights of a model with latent dangerous capabilities are made available to the public. Performing robust unlearning on a model whose weights are already publicly available is not helpful. Therefore we expect robust unlearning methods to be used by model developers after pretraining a model but before releasing it to the public. These developers would have access to the original pretraining dataset.
>
> The TOFU benchmark gives us another setup to test our method at scale. TOFU's fictional figures don't exist in any pretraining data, making the reference model mimic random initialization (zero latent knowledge of what we want to unlearn). We confirm our method works at larger scales and also test it across different architectures (Gemma-2-2B and Phi-1.5) and unlearning algorithms (GradDiff and NPO):
>
> **Table: TOFU Results on Gemma-2-2B with GradDiff**
>
> | Model | Initial Performance | Performance After 100-step Relearning |
> |-------|-------------------|----------------------------------|
> | **Unlearned Only** (GradDiff) | Forget Q&A: 0.000, MIA: 0.000 | Forget Q&A: 0.378, MIA: 0.822 |
> | **Unlearned+Distilled** (GradDiff+Distill) | Forget Q&A: 0.000, MIA: 0.011 | Forget Q&A: 0.293, MIA: 0.551 |
> | **Control** (Never saw TOFU) | Forget Q&A: 0.057, MIA: 0.310 | Forget Q&A: 0.269, MIA: 0.455 |
>
>
> **Table: TOFU Results on Phi-1.5 with NPO**
>
> | Model | Initial Performance | Performance After 100-step Relearning |
> |-------|-------------------|----------------------------------|
> | **Unlearned Only** (NPO) | Forget Q&A: 0.063, MIA: 0.186 | Forget Q&A: 0.375, MIA: 0.840 |
> | **Unlearned+Distilled** (NPO+Distill) | Forget Q&A: 0.063, MIA: 0.164 | Forget Q&A: 0.282, MIA: 0.692 |
> | **Control** (Never saw TOFU) | Forget Q&A: 0.131, MIA: 0.325 | Forget Q&A: 0.243, MIA: 0.465 |
>
> *Forget Q&A: Probability of correctly answering forget set author questions (lower is better)*
>
> *MIA: Membership Inference Attack success rate (lower is better)*
>
> Both our distilled models achieve relearning rates closer to the control baseline (a model that never saw the forget authors), while unlearned-only models rapidly recover the suppressed knowledge (trend is clearer in a graph). In the camera-ready version, we would report an analysis across model families and sizes.
>
> Importantly, we don't necessarily require starting from a random init model if we know for sure from which split of the pretraining data the forget target is from. Consider a case where there are several model checkpoints saved during training based on time intervals (m1, m2, m3, m4...) corresponding to dataset splits (d1, d2, d3, d4...), as is common for many production setups. If we know that the model has become problematic at a certain time point (e.g., harmful content introduced in d4), we can roll back to checkpoint m3 (which doesn’t have latent traces of d4 knowledge) and distill the unlearned m4 onto it. .
>
> ---
>
> ### W2: Explanation for why distillation robustifies unlearning
>
> Thank you for this comment *“The paper provides strong empirical evidence that distillation robustifies unlearning … A deeper analysis … could further strengthen the paper's contribution”*.
>
> We're particularly excited that our work aligns with emerging insights in the unlearning literature. Recent work [1] found that irreversible unlearning is difficult to achieve without "large rotations of principal directions, centroid shifts, and vanishing Fisher mass across many layers," essentially requiring many layers to undergo coordinated, large-magnitude perturbations. We believe our work provides a timely contribution by demonstrating a concrete mechanism that achieves what these works hypothesize as true forgetting.
>
> Our intuition is that all knowledge must pass through the teacher's outputs to reach the student. When combined with a student that initially has no latent information of the forget set, this distillation operation creates the kind of coordinated, large-magnitude perturbations across many layers that [1] identifies as necessary for irreversible unlearning. By starting from random initialization (or a clean checkpoint), the student undergoes global representational shifts while selectively learning only what the teacher demonstrates.
>
> Our experiments validate this hypothesis. Even when distilling on the full dataset including forget data:
>
> **Table: Selective Learning During Distillation**
>
> | Domain | Metric | Initial → Final |
> |--------|--------|-----------------|
> | Language (Retain: English) | CE Loss | 12.52 → 4.98 |
> | Language (Forget: Korean) | CE Loss | 12.52 → 12.64 |
> | Arithmetic (Retain: Add/Sub) | Accuracy | 0% → 86% |
> | Arithmetic (Forget: Mul/Div) | Accuracy | 0% → 0% |
>
> The student's performance on forget domains remains at random initialization levels despite seeing the forget domain data in every batch.
>
> Also, our approach addresses a broader category of capability removal beyond other privacy-focused unlearning use cases. While privacy applications typically target specific data points or narrow facts, our work demonstrates removal of entire capability domains (languages, mathematical operations).
>
> ---
>
> ### Q1: Preliminary results for wider variety of model families and larger sizes?
>
> We will prepare a wider variety of model families and larger sizes results for TOFU.
>
> Our main language and arithmetic unlearning experiments were done at pretraining scale, making it harder to scale up. We have initial results for up to 0.9B establishing similar findings, but for the purpose of focused scientific contribution, we chose to present singular model sizes (0.1B for language and 0.3B for arithmetic). Instead, we take an extremely strong adversarial assumption to ensure the validity and replicability of our claims, where we do hyperparameter search across 10 learning rates (1e-6 to 1e-3), substantial data (8M tokens for Korean, 2M tokens for arithmetic), and 500 gradient updates, and report adversarially optimal cases.
>
> ---
>
> ### Q2: Have you tried latent knowledge probes rather than behavioral scores?
>
> We have tested prompt-based and activation steering elicitation methods on our bilingual models. However, it was challenging to get clear signals, potentially due to the relatively small model sizes we used (100M-300M parameters) or limitations in applying these techniques to our specific setups. However, we tested alternative elicitation methods. For quantization-based attacks [2]:
>
> **Table: INT4 Quantization Robustness (TOFU, Gemma-2-2B)**
>
> | Method | Forget Q&A Prob | Forget Truth Ratio | Model Utility |
> |--------|----------------|-------------------|---------------|
> | GradDiff (FP16) | 0.000 | 0.000 | 0.495 |
> | GradDiff (INT4) | 0.008 | 0.185 | 0.242 |
> | GradDiff+Distilled (INT4) | 0.003 | 0.017 | 0.391 |
>
> Our distilled models show much better resilience across all metrics, suggesting that distillation provides defense against attack vectors beyond just gradient-based relearning.
>
> ---
>
> ### Q3: Performance drop on MMLU - how to mitigate without original pretraining data?
>
> See our comment above regarding threat models. Also, we think parts of our distillation setup could be optimized (e.g., spreading out the noising over distillation may help).
>
> ---
>
> ### Q4: Concrete quantitative comparison of cost?
>
> While Figure 5 shows that arithmetic reaches gold standard robustness with about 60% of gold standard compute and language with about 80%, we concretely mean data efficiency. The key advantage is avoiding the need to label the entire pretraining corpus. In Figures 5 and 7, UNDO required only 0.01% of the pretraining data to be labeled to match the gold standard robustness of full data filtering. That is, we create a new solution space:
>
> | Method | Robustness | Data Labeling |
> |-|-|-|
> | Data Filtering | High | High |
> | Standard Unlearning | Low | Low |
> | **Unlearn + Distill** | **High** | **Low** |
>
> ---
>
> ### Q5: Alternative methods of "damaging" the teacher?
>
> We see several promising alternatives like pruning, targeted damaging in ROME-style [3]. While we haven't implemented these yet, even imperfect targeting could improve retain-forget tradeoffs compared to our uniform approach.
>
> ---
>
> ### Q6: Task-dependent alpha - why does it vary? Practitioner guidance?
>
> The variation could stem from different evaluation metrics (cross-entropy loss for language vs. accuracy for arithmetic) having different sensitivities to perturbation. After an inflection point, we observe a linear trend between compute requirements and robustness in both domains once the noising starts to remove the latent traces.
>
> ---
>
> ### Q7: Broken references
>
> Thank you for catching this! We'll fix all references showing as "??" in the camera-ready.
>
> ---
>
> — We thank the reviewer for a very thorough review! —
>
> [1] Unlearning Isn't Deletion: Investigating Reversibility of Machine Unlearning in LLMs. Preprint 25
>
> [2] Catastrophic Failure of LLM Unlearning via Quantization. ICLR 25
>
> [3] “Locating and Editing Factual Associations in GPT” NeurIPS 22

---

> > ### Comment · Reviewer_TRF9 · 2025-08-04
> >
> > Thank you for your thoughtful response and the improvements you’ve proposed. I will retain my current score.

---

### Official Review · Reviewer_4DBt · 2025-07-02

**Clarity:** 4
**Significance:** 3
**Originality:** 3
**Rating:** 5
**Confidence:** 3

**Summary:**

This paper proposes a way to improve robustness of machine unlearning methods (which aim to remove certain harmful knowledge/capabilities). The problem: current unlearning methods produce models that can still elicit the harmful behavior on finetuning. The paper presents evidence of this problem in Sec 3.

The paper then proposes a way to improve the robustness of any existing unlearning method (i.e. plug-and-play), at the cost of additional compute. The key insight is to use distillation on unlearned models. This method is called Unlearn-and-Distill, presented in Sec 4.

The paper introduces a way to control the trade-off between additional compute and unlearning robustness. This method is called UNDO (Unlearn-Noise-Distill-on-Outputs), presented in Sec 5. It also compares with other unlearning methods in Sec 6.

**Questions:**

**Q1**. In the formulation in Section 2, the setup requires a retain set. Can unlearning work without a retain set, and are there algorithms in the literature? I ask this because practically, one might want to retain all capabilities except a few harmful ones. Thinking ahead, in this case, can distillation provide an advantage on existing unlearning methods?

**Q2**. W.r.t lines 169-173, what is the dataset used for distillation? Is it the full training dataset?

**Q3**. W.r.t lines 204-206, I'm unclear about whether this claim has been shown empirically, because your approach (line 100 and around) requires the entire training set to be partitioned into the retain and forget sets.

**Ethical Concerns:**

["NO or VERY MINOR ethics concerns only"]

**Final Justification:**

My final rating is 5 (increased from the initial rating of 4).

The initial rating of 4 was because:
- The paper makes an intuitive yet interesting observation,
- and it shows the idea works through detailed experiments.

My rating has increased because:
- I find the discussion by the authors on the weaknesses raised (W1 and W2) satisfactory.

**Limitations:**

Yes.

**Paper Formatting Concerns:**

No concerns.

**Quality:**

3

**Strengths And Weaknesses:**

**Strengths**

**S1**. The paper makes an interesting and novel observation that distillation robustifies unlearning. This approach, called Unlearn-and-Distill, gives advantage over simple unlearn baselines (Figure 4).

**S2**. The paper introduces an approach which allows for trading between compute and robustness (UNDO, Figure 5).

**S3**. The paper demonstrates that UNDO extends the pareto-optimal frontier on the tradeoff between quality retention (for harmless cases) and unlearning (for harmful cases). Figures 7 and 8 demonstrate competitive performance of UNDO over baselines.

**S4**. The paper is well written: It has a good structure that follow nicely, and explanations are clear to understand.

**Weaknesses**

The paper does not present much intuition/explanation of why distillation is effective at improving the robustness of unlearning methods. At a high-level, distillation seems to perform unlearning + "mixing" the unlearned weights with noise. Are there ways to investigate why this works?

Relatedly, are there other ways to "mix" the unlearned weights with noise instead of distillation? Perhaps they can be cheaper in compute?

---

> ### Author Rebuttal · Authors · 2025-07-31
>
> We thank the reviewer for the thoughtful review and for highlighting the novelty of our approach!
>
> ---
>
> ### Conceptual explanation for our work
>
> We appreciate your request for more intuition about why distillation works. While we give this intuition in Section 3, we've added some discussion here.
>
> To briefly recap our key finding from Section 3: We started with models that already possessed harmful capabilities (we refer to these as "reference models," which were pretrained on both English and Korean). We then trained these contaminated models to perfectly mimic an oracle model that had never seen Korean. Despite achieving near-perfect behavioral alignment, something strange happened during relearning attacks.
>
> When we compared three types of models:
>
> 1. **Oracle**: Never saw Korean during training
>
> 2. **Student (Random)**: Fresh model trained to mimic the oracle's outputs
>
> 3. **Student (Reference)**: Contaminated model trained to mimic the oracle's outputs
>
> The contaminated models that "forgot" Korean relearned it dramatically faster:
>
> **Table: Oracle Matching Results (Figure 2)**
> | Task | Model Type | Starting Knowledge | Relearned (100 steps) | Relearned (500 steps)
> |------|------------|------------------|------------------------------|------------------------------
> | Language | Oracle Teacher | English only | Korean CE: 5.8 | Korean CE: 4.2
> | Language | Student (Random) | None (fresh start) | Korean CE: 5.7 | Korean CE: 4.2
> | Language | Student (Reference) | English + Korean | Korean CE: **3.9** | Korean CE: **3.7**
> | Arithmetic | Oracle Teacher | Add/Sub only | Accuracy: 17% | Accuracy: 45%
> | Arithmetic | Student (Random) | None (fresh start) | Accuracy: 18% | Accuracy: 45%
> | Arithmetic | Student (Reference) | All operations | Accuracy: **45%** |Accuracy: **85%**
>
> Student (Reference) showed fast relearning despite behaving perfectly the same as the oracle.
>
> The model's "origin" (what it knew before matching the oracle) determined its vulnerability to relearning.
>
> Now, what if we could combine unlearning (which achieves behavioral suppression) with distillation to a fresh model (which has no problematic history)? When distilling from an unlearned teacher to a fresh student:
>
> 1. The student starts clean (no pre-existing harmful knowledge)
>
> 2. The teacher's outputs on harmful content are meaningless noise
>
> 3. The student can only learn from what the teacher demonstrates
>
> 4. Since the teacher refuses to teach harmful content, it cannot transfer
>
> That is, we leverage distillation as a capability filter and create a new solution space:
>
> | Method | Robustness | Data Labeling Required |
> |--------|------------|----------------------|
> | Data Filtering | High | Must label entire corpus |
> | Standard Unlearning | Low | Label <0.01% of data |
> | **Unlearning + Distillation** | **High** | **Label <0.01% of data** |
>
> We also validated this on TOFU [1], where fictional authors were created to test unlearning in realistic scenarios. Our distilled models relearned at similar speeds as the control baseline (a model that never saw the forget authors), while unlearned-only models rapidly recovered the suppressed knowledge under relearning attacks.
>
> Given that Section 3 already presents this conceptual foundation with evidence (Figure 2), would more forward references from Section 4 to Section 3 help clarify the connection in the manuscript?
>
> ---
>
> ### W1: Why does distillation robustify unlearning?
>
> *From review text: “The paper does not present much intuition/explanation of why distillation is effective at improving the robustness of unlearning methods.”*
>
> The student starts knowing nothing and can only learn what the teacher demonstrates. Since the unlearned teacher produces meaningless outputs on forget data, there's nothing for the student to learn about those capabilities.
>
> Consider a teacher that has undergone unlearning to suppress multiplication. Although latent multiplication knowledge remains in its parameters (which might later promote rapid relearning of the capability), when asked to solve multiplication problems, it produces only random noise. A fresh student can only learn from the gradients provided by the teacher's outputs and cannot acquire multiplication, because the teacher provides no meaningful signal.  Though intuitive once stated, the effectiveness of this approach is not obvious, given potential anomalies in the long tail of teacher logits. Our contribution is demonstrating that it works well in practice.
>
> As an example, the table below shows how the Korean CE Loss stays large despite the fact that the distillation training set includes Korean documents:
>
> *Table: Distillation - Language Unlearning Experiment*
>
> | Step | Student’s Eng CE Loss (Retain) | Student’s Kor CE Loss (Forget)
> |------|----------------|----------------
> | 0    | 12.52          | 12.52
> | 50   | 7.17           | 12.48
> | 200  | 6.02           | 12.58
> | 800  | 4.98           | 12.64
>
> Similarly, in arithmetic experiments, the student recovers the teacher's addition/subtraction/English performance while multiplication/division stays completely unlearned.
>
> *Table: Distillation - Arithmetic Unlearning Experiment*
>
> | Step | Student’s Eng CE Loss (Control) | Student’s Subtraction Acc (Retain) | Student’s Addition Acc (Retain) | Student’s Multiplication Acc (Forget) | Student’s Division Acc (Forget)
> |------|----------------|-----------------------|-------------------|----------------|----------
> | 1    | 12.60          | 0%               | 0%            | 0%             | 0%
> | 200  | 5.59           | 7%              | 3%            | 0%             | 0%
> | 400  | 4.76           | 18%             | 3%            | 0%             | 0%
> | 1000 | 4.47          |92%        |80%     | 0%         | 0%
>
>
>
> ---
>
> ### W2: Alternative approaches to mixing
>
> *From review text: “Relatedly, are there other ways to "mix" the unlearned weights with noise instead of distillation?”*
>
> We believe that this is a promising topic for future research. Instead of random perturbations, we could use interpretability to identify and corrupt specific weights responsible for harmful behaviors. We think success would depend on two criteria:
>
> Criterion 1: Size of unlearning target - How semantically broad the capability is
>
> Criterion 2: Entanglement - How distributed across the model's weights
>
> For narrow facts like "the capital of France is Paris" (small C1), mechanistic interpretability suggests these are often localized (small C2) [2], making targeted noising promising. However, for complex capabilities like deception (large C1), the required computations are likely distributed across many components (large C2), such as tracking truth, generating alternatives, modeling beliefs, and optimizing outputs [3].
>
> While we haven't implemented targeted noising yet, even imperfect targeting could improve retain-forget tradeoffs. It doesn't need to be perfect, just better than global. This is an exciting avenue for future work!
>
> ---
>
> ### Q1: Unlearning without explicit retain sets
>
> *From review text: “Can unlearning work without a retain set, and are there algorithms in the literature?”*
>
> Yes, this depends on the unlearning algorithm. GradDiff computes `loss_forget - loss_retain`, requiring both sets. In practice, many past works use generic retain datasets (WMDP uses Wikitext). During distillation, we don’t require an explicit retain set. The teacher's outputs on any unlabeled data automatically provide the training signal, where meaningful predictions guide learning on retain content while uninformative outputs prevent learning on forget content.
>
> ---
>
> ### Q2: Distillation dataset
>
> *From review text: “what is the dataset used for distillation? Is it the full training dataset?”*
>
> Yes, we use the full pretraining dataset for distillation. Only the initial unlearning step needs small labeled retain/forget sets (typically <0.01% of data). We avoid labeling billions of tokens.
>
> ---
>
> ### Q3: Full training set partitioning
>
> *From review text: “I'm unclear about whether this claim has been shown empirically, because your approach (line 100 and around) requires the entire training set to be partitioned into the retain and forget sets.”*
>
> No, we don't require partitioning the entire training set. Lines 204-206 compare against the theoretical "gold standard" of filtering everything (impractical at scale). We only need small labeled sets for initial unlearning, then use unlabeled pretraining data for distillation.
>
> ---
>
> — We thank the reviewer for pushing us to clarify the conceptual points! —
>
> [1] TOFU: A Task of Fictitious Unlearning for LLMs. COLM 24
>
> [2] Locating and Editing Factual Associations in GPT. NeurIPS 22
>
> [3] The Geometry of Truth: Emergent Linear Structure in Large Language Model Representations of True/False Datasets. COLM 24

---

> > ### Comment · Reviewer_4DBt · 2025-08-03
> > **Thanks for the rebuttal**
> >
> > Thank you for the rebuttal. Your responses to weaknesses W1 and W2 answers my questions. It would be great if you can include these points in the final manuscript. I believe this clearly explains the intuition behind distillation for unlearning + discusses potentially other ways to achieve the same result.

---

> ### Author Response · Authors · 2025-08-03
> **Response to the reviewer**
>
> Thank you for engaging with our rebuttal. We will incorporate the intuition discussion into the manuscript as suggested. Given that W1 and W2 were your primary concerns and these have been addressed, we wanted to confirm whether any aspects of the work remain unclear or require further clarification.

---

### Official Review · Reviewer_4Vbg · 2025-07-02

**Clarity:** 3
**Significance:** 3
**Originality:** 3
**Rating:** 4
**Confidence:** 3

**Summary:**

The paper presents an investigation into the limitations of current machine unlearning techniques, particularly targeting performance with respect to relearning attacks, where a model quickly recovers forgotten capabilities through finetuning. The authors argue that conventional unlearning often suppresses undesired behaviors rather than eliminating them, leaving latent representations intact due to factors like plasticity and representational entanglement. To address this, the authors propose Unlearn-and-Distill, a method where a model is first unlearned and then used to train a freshly initialized student model through knowledge distillation, effectively discarding residual capabilities. Building on this idea, they introduce UNDO (Unlearn-Noise-Distill-on-Outputs), a more flexible and computationally efficient extension.

The authors empirically validate their methods across multiple domains, including language modeling, arithmetic reasoning, and a high-stakes Weapons of Mass Destruction Proxy (WMDP) benchmark. The results show that UNDO shows robustness against relearning attacks compared to existing unlearning methods such as MaxEnt, SAM, and RepNoise. Experiments demonstrate that increasing the noise perturbation parameter α consistently strengthens resistance to relearning, with even modest values yielding improvements. Moreover, the paper shows that UNDO achieves favorable retain-forget trade-offs and remains resilient under strong adversarial finetuning (e.g., 500-step attacks, also highlighted in Figure 7).

**Questions:**

* While the paper emphasizes that the mixing coefficient $\alpha$ largely controls robustness and computation cost, the role of the noise scale $\beta$ seems less clearly justified. It would be good if the authors could provide more detailed insights for this parameter in practive. For instance, are there conditions or dataset properties under which certain beta values outperform others?

* The paper evaluates robustness under several adversarial relearning scenarios, but the extent and variety of these attacks could be expanded (and would never be exhaustive). This raises some questions regarding the guarantee of unlearning, which remains a challenge in general. It would be good if the authors could comment on similar lines, explaining the extent and limitations of the current setting.

* The experiments focus on retain and forget sets to measure unlearning effectiveness, but it is unclear how the method affects generalization to unrelated downstream tasks or out-of-distribution data. It would be good if the authors provide evaluations or discussions on whether the UNDO method preserves overall model utility and fairness across a wider task spectrum. Additionally, evaluation using richer and more comprehensive metrics such as Verbatim Probability, ROUGE variants, Model Utility, Forget Quality, TruthRatio, and Membership Inference Attack (MIA) strength, as used in recent benchmarks (e.g., Open-Unlearning), could further help validate the claims made in the paper.

**Ethical Concerns:**

["NO or VERY MINOR ethics concerns only"]

**Final Justification:**

The added set of experimental results further validates the claims and makes it more aligned with the unlearning literature. After considering the authors’ rebuttal and additional experiments, I am updating the rating by a point to reflect the improvements made. Below is a summary of resolved/remaining concerns:

The resolved concerns include 1) the addition of comprehensive evaluations on TOFU using open-unlearning benchmarks, directly addressing the concern about alignment with the broader unlearning community. 2) The inclusion of evaluation metrics that are popular in the unlearning community, Verbatim Probability, ROUGE variants, Model Utility, Forget Quality, TruthRatio, and MIA strength. 3) Authors also tested quantization-based elicitation and demonstrated improved robustness via post-distillation, further showing the generality of their approach. 3) The role of β is now better explained, with practical guidance that was added after the rebuttal.

Some of the remaining concerns include 1) the approach still relies on access to unfiltered data during distillation, which may not always be feasible (e.g., open-weight models without pretraining data). 2) While TOFU results are encouraging, broader adoption across more diverse or realistic forgetting scenarios (e.g., non-synthetic or semantically entangled data) would further strengthen the work.

After the rebuttal, the paper is above the acceptance threshold, based on the substantial improvements made and the added set of experiments.

**Limitations:**

Yes, the authors clearly mention some of the major limitations of their approach, particularly around the distillation and the challenges of applying UNDO in settings where access to the original pretraining data is limited (e.g., Gemma-2-2B). These limitations make the proposed approach difficult to apply to all the open-weight LLMs where pretraining data is available.

**Quality:**

3

**Strengths And Weaknesses:**

**Strengths**

* The paper systematically demonstrates that standard unlearning methods leave residual capabilities that can be quickly relearned (similar findings are also revealed in other recent works [1, 2, 3, 4]). Intriguingly, the paper introduces a controlled methodology (referred to as UNDO) that allows interpolation between noising and full reinitialization, with some empirical improvements in resisting relearning across multiple domains (language, arithmetic, WMDP), that may be helpful for future works on similar lines.

* While distillation has been used for compression or privacy, this work is somewhat novel in explicitly using it as a means to destroy/degrade/suppress latent capacities that contribute to fast relearning (making the idea itself unique and less explored in the unlearning community [though some initial works like [5] do explore a teacher-student approach]). This reframing is both conceptually and technically significant, as it connects behavioral unlearning with representation-level damage via noising, followed by reconstruction under behavioral constraints, making it a good unlearning mechanism for the community to explore in future works. Making it align with the unlearning community by using similar benchmarks would definitely benefit the overall quality/utility of this work.

**Weaknesses**

* One of the notable weaknesses of this work is that it evaluates robustness primarily through forget/retain set performance and relearning speed. However, the effects of perturbation and distillation on general downstream performance (e.g., OOD generalization, calibration, fairness) remain unexplored. Since the proposed UNDO method introduces large-scale noising and re-alignment, understanding its unintended side effects on broader capabilities becomes imperative. It would be good to add other evaluation metrics that are used in unlearning [also refer to questions for clarification details regarding other evaluation metrics]

* Another major weakness is that the paper uses a completely different setting than the existing works on machine unlearning, making it difficult to compare/validate/check how the proposed unlearning fits the existing set of works. It would be good to add some results on benchmarks like open-unlearning, using datasets like TOFU for a better alignment with the unlearning research community.

* On the method side, the current method distills only output distributions (via forward KL), potentially ignoring subtle representational changes. It remains unclear whether internal activations, gradients, or alignment signals are also preserved or transformed in ways that might make the model vulnerable to alternative elicitation strategies. Some works like [1, 2, 3, 4] show challenges in measuring unlearning and how the model’s knowledge remains intact even after unlearning. Since it is difficult to compare the proposed work using different datasets, it remains challenging to rely on the results for the applicability of the proposed learning method in other settings.

[1] https://arxiv.org/abs/2410.16454

[2] https://arxiv.org/abs/2406.13356

[3] https://arxiv.org/abs/2411.15477

[4] https://arxiv.org/abs/2402.16835

[5] https://arxiv.org/abs/2310.02238

---

> ### Author Rebuttal · Authors · 2025-07-31
>
> We thank the reviewer for the thoughtful feedback and constructive suggestions! First, we share additional evaluations on TOFU to address the concerns about downstream performance and alignment with the broader unlearning literature.
>
> ---
>
> ### TOFU
>
> We conducted experiments using the **open-unlearning** library. TOFU turns out to be ideal since its fictional figures don't exist in any pretraining data, making the base model mimic random initialization (never encountered what we want to unlearn). We have done additional experiments with Gemma 2 2B and Phi 1.5. We report Gemma 2 2B in this response due to character limits.
>
> *Step 1: TOFU Finetuning on gemma-2-2b*
>
> | Epoch | Forget_Q_A_Prob | Forget_ROUGE | Forget_Truth | Retain_Q_A_Prob | Retain_ROUGE | Model_Utility | Extraction_Str | MIA_MinK |
> |-------|-----------|--------------|--------------|-----------|--------------|---------|---------|------|
> | 1     | 0.29      | 0.42         | 0.72         | 0.29      | 0.40         | 0.35    | 0.13    | 0.53 |
> | 3     | 0.51      | 0.49         | 0.56         | 0.49      | 0.46         | 0.43    | 0.21    | 0.92 |
> | 5     | 0.59      | 0.53         | 0.48         | 0.57      | 0.49         | 0.45    | 0.24    | 0.97 |
>
> The model memorizes the fictional authors, with Forget_QA probability increasing from 0.06 (epoch 0) to 0.59, while MIA (membership inference attack) success rate goes from 0.31(epoch 0)  to 0.97.
>
> We then apply GradDiff unlearning, which successfully suppresses the unwanted knowledge while maintaining retain performance.
>
> *Step 2: Unlearning TOFU using GradDiff*
>
> | Step | Forget_Q_A_Prob | Forget_ROUGE | Forget_Truth | Retain_Q_A_Prob | Retain_ROUGE | Model_Utility | Extraction_Str | MIA_MinK |
> |------|-----------|--------------|--------------|-----------|--------------|---------|---------|-------|
> | 0    | 0.59      | 0.53         | 0.48         | 0.57      | 0.49         | 0.45    | 0.24    | 0.97  |
> | 25   | 0.04      | 0.36         | 0.49         | 0.31      | 0.41         | 0.46    | 0.04    | 0.37  |
> | 50   | 0.001    | 0.003        | 0.002        | 0.41      | 0.36         | 0.44    | 0.03    | 0.01  |
>
> After 50 steps, the forget metrics drop to near-zero (Forget_QA: 0.59→0.001) while model utility remains stable. We then distill this behaviorally-suppressed model back into the original gemma-2-2b:
>
> *Step 3: Distilling Unlearned Model*
>
> | Epoch | Forget_Q_A_Prob | Forget_ROUGE | Forget_Truth | Retain_Q_A_Prob | Retain_ROUGE | Model_Utility | Extraction_Str | MIA_MinK |
> |-------|-----------|--------------|--------------|-----------|--------------|---------|---------|-------|
> | 1     | 0.001     | 0.01         | 0.006        | 0.21      | 0.36         | 0.38    | 0.03    | 0.006 |
> | 3     | 0.001    | 0.002        | 0.001        | 0.26      | 0.37         | 0.44    | 0.03    | 0.01  |
> | 5     | 0.001    | 0.002        | 0.001        | 0.26      | 0.37         | 0.44    | 0.03    | 0.01  |
>
> The resulting rewritten model maintains near-zero forget metrics while recovering full utility (0.44), matching the unlearned teacher's performance. This demonstrates that distillation transfers desired behaviors on retain but not on forget.
>
> Then, we subject the three models (unlearned, unlearned+distilled, fresh base model - as control) to relearning attacks (100 steps on TOFU) to measure how quickly the forget knowledge is recovered:
>
> *Metric: Forget_Q_A_Prob*
>
> | Model Type | Step 10 | Step 20 | Step 50 | Step 100 |
> |------------|---------|---------|---------|----------|
> | (a) Unlearned | 0.313 | 0.331 | 0.362 | 0.378 |
> | (b) Unlearned+Distilled | 0.242 | 0.261 | 0.283 | 0.293 |
> | (c) Control (Never seen TOFU) | 0.226 | 0.242 | 0.261 | 0.269 |
>
> *Metric: Extraction Attack Success Rate*
>
> | Model Type | Step 10 | Step 20 | Step 50 | Step 100 |
> |------------|---------|---------|---------|----------|
> | (a) Unlearned | 0.137 | 0.148 | 0.154 | 0.155 |
> | (b) Unlearned+Distilled | 0.113 | 0.117 | 0.123 | 0.122 |
> | (c) Control (Never seen TOFU) | 0.114 | 0.119 | 0.122 | 0.124 |
>
> While the unlearned-only model rapidly recovers the suppressed knowledge (Forget_QA: 0.378 after 100 steps), the distilled model relearns at rates closer to the control, a fresh model that has never seen TOFU. Extraction attack rates are virtually identical between our method and the control (0.122 vs. 0.124), suggesting that the post-unlearning distillation step does promote a deeper knowledge removal. This pattern holds across different unlearning methods (GradDiff, NPO, etc) that are used before distillation, confirming the generality of our approach.
>
> ---
>
> ### W1: Effects on general performance
>
> TOFU results above directly demonstrate capabilities preservation during specific knowledge removal. The distilled model maintains the same utility as the teacher while keeping forget metrics near zero. Our controlled experiments show the student selectively learns general capabilities that the unlearned teacher properly demonstrates, despite distilling on the full, unfiltered dataset:
>
> *Table: Distillation Step - Language Unlearning*
>
> | Step | Student's Eng CE Loss (General) | Student's Kor CE Loss (Forget) |
> |------|----------------|----------------|
> | 0    | 12.52          | 12.52         |
> | 50   | 7.17           | 12.48         |
> | 200  | 6.02           | 12.58         |
> | 800  | 4.98           | 12.64         |
>
> Similarly, in arithmetic experiments, the student recovers the teacher's addition/subtraction/English performance while multiplication/division stays completely unlearned.
>
> *Table: Distillation Step - Arithmetic Unlearning*
>
> | Step | Student's Eng CE Loss (Control) | Student's Subtraction Acc (Retain) | Student's Addition Acc (Retain) | Student's Multiplication Acc (Forget) | Student's Division Acc (Forget) |
> |------|----------------|-----------------------|-------------------|----------------|----------|
> | 1    | 12.60          | 0%               | 0%            | 0%             | 0%       |
> | 200  | 5.59           | 7%              | 3%            | 0%             | 0%       |
> | 400  | 4.76           | 18%             | 3%            | 0%             | 0%      |
> | 1000 | 4.47          | 92%        | 80%     | 0%         | 0%   |
>
> These results are also available as Figure 9. In all experiments reported (Figures 1-6), the distilled student models achieve performance within 5% of their teachers on English \& retain task. For certain tasks like arithmetic, we observe that the student actually often exceeds the teacher's performance on retain and control tasks. This is reminiscent of a well-documented phenomenon in distillation literature where students can surpass their teachers [1]. Distillation not only preserves capabilities but can also sometimes enhance them through cleaner learned representations.
>
>
> ---
>
> ### W2: Validation on mainstream benchmarks
>
> The TOFU experiments above address this concern. We tested multiple unlearning methods from open-unlearning, and adding distillation consistently improved resistance to relearning.
>
> We thank the reviewer for this suggestion. Our method validated well on mainstream benchmarks like TOFU. Given these compelling results, we will:
>
> 1. Contribute our method to **open-unlearning** for ease of access.
>
> 2. Add a separate TOFU evaluation section.
>
> ---
>
> ### W3: Other elicitation efforts
>
> Thank you for raising this point about alternative elicitation strategies!
>
> We first note that prior literature shows that finetuning is a strong elicitation technique. [2, 3] found that while finetuning consistently performed best across all tested setups, other attacks showed limited success. This universal vulnerability to finetuning, combined with its standardized implementation and replicability, motivated our focus on finetuning.
>
> In our paper, we go a step further and take the strongest adversarial assumption of hyperparameter search across 10 learning rates (1e-6 to 1e-3), substantial data (8M tokens for Korean, 2M tokens for arithmetic), 500 gradient updates, and report adversarially optimal cases.
>
> Additionally, we empirically verify that our approach also robustifies against alternative elicitation strategies. We tested quantization using the open-unlearning library [4]:
>
> *Table: Quantization Effects*
>
> | Method | Metric | Original | Quantized |
> |--------|--------|------|------|
> | Unlearned (GradDiff) | forget_Q_A_Prob | 0 | 0.008 |
> | | forget_truth_ratio | 0 | 0.185 |
> | | model_utility | 0.495 | 0.242 |
> | | mia_min_k | 0.0004 | 0.054 |
> | Unlearned+Distilled | forget_Q_A_Prob | 0 | 0.003 |
> | | forget_truth_ratio | 0.0006 | 0.017 |
> | | model_utility | 0.442 | 0.391 |
> | | mia_min_k | 0.011 | 0.013 |
>
> Under INT4 quantization, standard unlearning shows increased forget metrics (forget_Q_A_Prob: 0 → 0.008, forget_truth_ratio: 0 → 0.185). Our distilled models demonstrate superior robustness with forget_Q_A_Prob reaching only 0.003 (2.7× better) and forget_truth_ratio at 0.017 (11× better). The distilled models also maintain better utility (0.391 vs 0.242) and lower MIA vulnerability (0.013 vs 0.054).
>
> ---
>
> ### Q: Role of β
>
> We have edited our manuscript to clarify the role of β and include an explanation here. The role of β is to ensure sufficient perturbation at low α values (scaling to be meaningfully different from Xavier init) and also that the resulting model has the correct scale. We recommend: (1) Use β=0.1 as default (2) Only tune β for α<0.2 (3) For α≥0.3, the mixing coefficient provides sufficient disruption. This aligns with [5] who found β=0.01-0.1 optimal across various settings.
>
> ---
>
> — We thank the reviewer for the constructive suggestions! —
>
> [1] Understanding the gains from repeated self-distillation. NeurIPS 24
>
> [2] The elicitation game: Evaluating capability elicitation techniques. ICML 25
>
> [3] Eight methods to evaluate robust unlearning in llms. Preprint
>
> [4] Catastrophic Failure of LLM Unlearning via Quantization. ICLR 25
>
> [5] On warm-starting neural network training. NeurIPS 20

---

> ### Author Response · Authors · 2025-08-05
> **Response to Reviewer**
>
> We wanted to follow up on our rebuttal, as we haven't heard back, and the discussion period ends on August 8. We conducted additional experiments to address your main concerns:
>
> - Added TOFU experiments on Gemma-2-2B and Phi-1.5 for better alignment with the unlearning community
>
> - Evaluated additional metrics you suggested (Verbatim Probability, ROUGE variants, Model Utility, Forget Quality, TruthRatio, MIA strength)
>
> - Tested robustness to alternative elicitation strategies
>
> We'd greatly appreciate your thoughts on whether these additions address your concerns about limited evaluation and alignment with existing unlearning work. Are there any remaining issues we could clarify?

---

> > ### Comment · Reviewer_4Vbg · 2025-08-06
> >
> > Thank you for the detailed and thorough rebuttal with an additional set of experiments (sorry if it took too long to respond, the detailed rebuttal for all the reviewers took too much time to read). I am glad that the authors added the TOFU experiments, expanded metrics, and robustness evaluations (in such a short time). These additions clearly increase the quality of the paper and improve its alignment with the broader unlearning literature. I have taken these updates into consideration during the final evaluation and updated the scores accordingly (with justifications provided). Thank you again for the detailed clarifications and the added set of experiments.

---

> > > ### Author Response · Authors · 2025-08-07
> > > **Response to Reviewer**
> > >
> > > Thank you for reviewing our rebuttal and the additional experimental work. Your feedback throughout this process has been valuable in strengthening the paper's experiments. We appreciate that the TOFU experiments and expanded metrics helped clarify the method's position within the unlearning literature. The suggestions regarding benchmark alignment and evaluation scope led to meaningful improvements in the work. Thank you again for your suggestions!

---

### Note · Authors · 2025-08-13

We thank the Reviewers, ACs, and SACs for their engagement in the review process. We appreciate the recognition of our contributions and the valuable feedback that has strengthened our work:

- *"The paper tackles a highly significant and timely problem in LLM safety. [...] The experimental evaluation is exceptionally thorough and well-designed. [...] The paper is very well-written, clearly structured, and easy to follow"* **(Reviewer TRF9)**

- *"The paper makes an interesting and novel observation that distillation robustifies unlearning. [...] The paper is well written: It has a good structure that follow nicely, and explanations are clear to understand"* **(Reviewer 4DBt)**

- *"UNDO consistently sits on the pareto frontier of the utility-unlearning curve, [...]"* **(Reviewer nuYc)**

- *"While distillation has been used for compression or privacy, this work is somewhat novel in explicitly using it as a means to destroy/degrade/suppress latent capacities [...] This reframing is both conceptually and technically significant [...]"* **(Reviewer 4Vbg)**

The reviewers have emphasized several key strengths. Notably, the reviewers mentioned **novel core insight about distillation**, **thorough experimental validation**, and **clear practical implications**. Through our discussions with reviewers, we have addressed several key technical questions. We highlight a few here:

- **Alignment with broader community**: We conducted TOFU experiments on Gemma-2-2B and Phi-1.5. Our unlearned-and-distilled models show relearning rates comparable to control baselines (Forget_Q_A_Prob: 0.293 vs 0.269), while unlearned-only models rapidly recover forget capabilities compared to control baselines (Forget_Q_A_Prob: 0.378 vs 0.269), validating our approach on mainstream benchmarks.

- **Conceptual clarification**: We clarified that distillation acts as a capability filter since the student starts from random init and can only learn from the teacher's behavior. Our experiments confirm that forget domain performance remain at random init levels despite exposure to forget data in every batch.

- **Robustness beyond relearning attacks**: We demonstrated resilience against quantization attacks, with unlearned-and-distilled models showing 11x better resistance on forget truth ratio (0.017 vs 0.185 under INT4), addressing concerns about alternative elicitation strategies.

We appreciate reviewers for acknowledging these improvements.

Thank you again for the constructive dialogue.

---

### Decision · Program_Chairs · 2025-09-17

**Decision:**

Accept (spotlight)

**Comment:**

This paper addresses a critical robustness problem in machine unlearning for large language models. The authors make the key observation that distilling an unlearned model into a freshly initialized student robustifies the forgetting process, and further propose UNDO (Unlearn-Noise-Distill-on-Outputs) as a practical extension that interpolates between noising and full reinitialization.

Reviewers broadly praised the core insight, noting that using distillation as a capability filter is a technically and conceptually novel contribution. The evaluation is thorough, with additional experiments during rebuttal. For improvement, the main limitations concern (1) reliance on access to pretraining data for effective distillation, which restricts applicability to settings where data is not available, and (2) limited explanation of why distillation achieves robust unlearning. After the rebuttal, reviewers updated their scores positively.

The AC concurs with the reviewers that the overall strengths of the work clearly outweigh its weaknesses, and therefore recommends acceptance.